# Batch Bayesian optimisation via density-ratio estimation with guarantees

**Rafael Oliveira**[1,2*]
rafael.oliveira@sydney.edu.au

**Louis C. Tiao**[3]
louis.tiao@sydney.edu.au

**Fabio Ramos**[3,4]
fabio.ramos@sydney.edu.au

[1]Brain and Mind Centre, the University of Sydney, Australia
[2]ARC Training Centre in Data Analytics for Resources and Environments, Australia
[3]School of Computer Science, the University of Sydney, Australia
[4]NVIDIA, USA

## Abstract

Bayesian optimisation (BO) algorithms have shown remarkable success in applications involving expensive black-box functions. Traditionally BO has been set as a sequential decision-making process which estimates the utility of query points via an acquisition function and a prior over functions, such as a Gaussian process. Recently, however, a reformulation of BO via density-ratio estimation (BORE) allowed reinterpreting the acquisition function as a probabilistic binary classifier, removing the need for an explicit prior over functions and increasing scalability. In this paper, we present a theoretical analysis of BORE's regret and an extension of the algorithm with improved uncertainty estimates. We also show that BORE can be naturally extended to a batch optimisation setting by recasting the problem as approximate Bayesian inference. The resulting algorithms come equipped with theoretical performance guarantees and are assessed against other batch and sequential BO baselines in a series of experiments.

## 1   Introduction

Bayesian optimisation (BO) algorithms provide flexible black-box optimisers for problems involving functions which are noisy or expensive to evaluate [1]. Typical BO approaches place a probabilistic model over the objective function which is updated with every new observation in a sequential decision-making process. Most methods are based on Gaussian process (GP) surrogates [2], which provide closed-form analytic expressions for the model's posterior distribution and allow for a number of theoretical performance guarantees [3–5]. However, GP surrogates have a number of limitations, such as not easily scaling to high-dimensional domains, high computational complexity and requiring a careful choice of covariance function and hyper-parameters [2]. Non-GP-based BO methods have also been proposed in the literature, such as BO methods based on neural networks [6, 7] and random forests [8] regression models.

As an alternative to improving the model, Tiao et al. [9] focus on the acquisition function, which in BO frameworks represents the guide that takes the model predictions into account. They show that one can derive the acquisition function directly without an implicit model by reinterpreting

---

[*]Corresponding author.

the expected improvement [4, 1] via a density-ratio estimation problem. Applying this perspective, the acquisition function can then be derived as a classification model, which can be represented by flexible parametric models, such as deep neural networks, and efficiently trained via stochastic gradient descent. The resulting method, called *Bayesian optimisation via density-ratio estimation* (BORE) is then shown to outperform a variety of traditional GP-based and non-GP baselines.

Despite the significant performance gains, BORE has only been applied to a sequential setting and not much is known about the method's theoretical guarantees. Batch BO methods have the potential to speed up optimisation in settings where multiple queries to the objective function can be evaluated simultaneously [10–13]. Given its flexibility to apply models which can scale to large datasets, it is therefore a natural question as to whether BORE can be readily extended to the batch setting in a computationally efficient way.

In this paper, we extend the BORE framework to the batch setting and analyse its theoretical performance. To derive theoretical guarantees, we first show that the original BORE can be improved by accounting for uncertainty in the classifier's predictions. We then propose a novel method, called BORE++, which uses an upper confidence bound over the classifier's predictions as its acquisition function. The method comes equipped with guarantees in the probabilistic least-squares setting. We provide extensions for both BORE and BORE++ to the batch setting. Lastly, we present experimental results demonstrating the performance of the proposed algorithms in practical optimisation problems.

## 2 Background

We consider a global optimisation problem over a compact search space $\mathcal{X} \subset \mathbb{R}^d$ of the form:

$$\mathbf{x}^* \in \underset{\mathbf{x} \in \mathcal{X}}{\arg\min} \, f(\mathbf{x}) \,, \tag{1}$$

where $f : \mathcal{X} \to \mathbb{R}$ is assumed to be a black-box objective function, i.e., we have no access to gradients nor analytic formulations of it. In addition, we are only allowed to run up to $T$ rounds of function evaluations, where we might collect single points or batches of observations $y_t := f(\mathbf{x}_t) + \epsilon_t$, which are corrupted by additive noise $\epsilon_t$, for $t \in \{1, \dots, T\}$.

### 2.1 Bayesian optimisation

Bayesian optimisation (BO) algorithms approach the problem in Equation 1 via sequential decision making [1]. At each iteration, BO selects a query point by maximising an acquisition function $a$:

$$\mathbf{x}_t \in \underset{\mathbf{x} \in \mathcal{X}}{\arg\max} \, a(\mathbf{x}|\mathcal{D}_{t-1}) \tag{2}$$

The acquisition function encodes information provided by the observations collected so far $\mathcal{D}_t := \{\mathbf{x}_i, y_i\}_{i=1}^{t-1}$ using a probabilistic model over $f$, typically a Gaussian process (GP) [2], conditioned on the data. After collecting an observation $y_t$, the dataset is updated with the new query-observation pair $\mathcal{D}_t := \mathcal{D}_{t-1} \cup \mathbf{x}_t, y_t$. This process then repeats for a given number of iterations $T$.

### 2.2 Bayesian optimisation via density-ratio estimation (BORE)

The expected improvement (EI) [4, 14] is a popular acquisition function in the BO literature and the basis for many BO algorithms. At each iteration $t \geq 1$, one can define $\tau := \min_{i<t} y_t$ as an incumbent target. EI is then defined as:

$$a_{\mathrm{EI}}(\mathbf{x}|\mathcal{D}_{t-1}) := \mathbb{E}[\max\{0, \tau - f(\mathbf{x})\}|\mathcal{D}_{t-1}] \,. \tag{3}$$

In the case of a GP prior on $f|\mathcal{D}_{t-1} \sim \mathcal{GP}(\mu_{t-1}, k_{t-1})$, the EI is available in closed form as a function of the GP posterior. However, the EI may be reformulated without the need for a prior.

Under mild assumptions, Bergstra et al. [15] showed that the EI can be formulated as a density ratio between two probability distributions. Let $\ell(\mathbf{x}) := p(\mathbf{x}|y \leq \tau)$ represent the probability density over $\mathbf{x} \in \mathcal{X}$ conditioned on the observation $y$ being below a threshold $\tau \in \mathbb{R}$. Conversely, let $g(\mathbf{x}) := p(\mathbf{x}|y > \tau)$. For $\gamma \in [0, 1]$, the $\gamma$-relative density ratio between these two densities is:

$$\rho_\gamma(\mathbf{x}) := \frac{\ell(\mathbf{x})}{\gamma\ell(\mathbf{x}) + (1-\gamma)g(\mathbf{x})} \,, \quad \mathbf{x} \in \mathcal{X} \,, \tag{4}$$

**Algorithm 1:** BORE

---

**1 for** $t \in \{1, \dots, T\}$ **do**

**2** $\quad \tau := \hat{\Phi}_{t-1}^{-1}(\gamma)$

**3** $\quad z_i := \mathbb{I}[y_i \leq \tau], \quad i \in \{1, \dots, t-1\}$

**4** $\quad \tilde{\mathcal{D}}_{t-1} := \{\mathbf{x}_i, z_i\}_{i=1}^{t-1}$

**5** $\quad \hat{\pi}_t \in \operatorname{argmin}_\pi \mathcal{L}[\pi | \tilde{\mathcal{D}}_{t-1}]$

**6** $\quad \mathbf{x}_t \in \operatorname{argmax}_{\mathbf{x} \in \mathcal{X}} \hat{\pi}_{t-1}(\mathbf{x})$

**7** $\quad y_t := f(\mathbf{x}_t) + \epsilon_t$

**8 end**

---

noting that $\gamma = 0$ leads to the ordinary probability density ratio definition, $\rho_0(\mathbf{x}) = \ell(\mathbf{x})/g(\mathbf{x})$. Now if we choose $\tau := \Phi^{-1}(\gamma)$, where $\Phi(s) := p(y \leq s)$ represents the cumulative distribution function of the marginal distribution of observations,[2] for $s \in \mathbb{R}$, and then replace $\tau$ in Equation 3, Bergstra et al. [15] have shown that[3] $a_{\mathrm{EI}}(\mathbf{x}) \propto \rho_\gamma(\mathbf{x})$, for $\mathbf{x} \in \mathcal{X}$. Based on this fact, Tiao et al. [9] showed:

$$a_{\mathrm{EI}}(\mathbf{x}) \propto \rho_\gamma(\mathbf{x}) = \gamma^{-1}\pi(\mathbf{x}), \quad \mathbf{x} \in \mathcal{X}, \tag{5}$$

where $\pi(\mathbf{x}) := p(y \leq \tau | \mathbf{x})$ can be approximated by a probabilistic classifier trained with a proper scoring rule, such as the binary cross-entropy loss:

$$\mathcal{L}_t[\pi] := \sum_{i=1}^t z_i \log \pi(\mathbf{x}_i) + (1 - z_i) \log(1 - \pi(\mathbf{x}_i)). \tag{6}$$

Other examples of proper scoring rules include the least-squares loss, which leads to probabilistic least-squares classifiers [16], and the zero-one loss. We refer the reader to Gneiting and Raftery [17] for a review and theoretical analysis on this topic.

BORE is summarised in Algorithm 1. As seen, the marginal observations distribution CDF $\Phi(s) := p(y \leq s)$ is replaced by the empirical approximation $\hat{\Phi}_t(s) := \frac{1}{t} \sum_{i=1}^t \mathbb{I}[y_i \leq s]$ and its corresponding quantile function $\hat{\Phi}_t^{-1}$. At each iteration, observations are labelled according to the estimated $\gamma$th quantile $\tau$, and a classifier $\hat{\pi}_t$ is trained by minimising the loss $\mathcal{L}[\pi | \tilde{\mathcal{D}}_t]$ over the data points $\tilde{\mathcal{D}}_t$. A query point $\mathbf{x}_t$ is chosen by maximising the classifier's probabilities, which in our case corresponds to maximising the expected improvement. A new observation is collected, and the algorithm continues running up to a given number of iterations $T$. As demonstrated, no explicit probabilistic model for $f$ is needed, only a classifier, which can be efficiently trained via, e.g., stochastic gradient descent.

## 3 Analysis of the BORE framework

In this section, we analyse limitations of the BORE framework in modelling uncertainty and analyse its effects on the algorithm's performance. As presented in Section 2.2, at each iteration $t \geq 1$, the original BORE framework trains a probabilistic classifier $\hat{\pi}_t(\mathbf{x})$ to approximate $p(y \leq \tau | \mathbf{x})$, where $\tau$ denotes the $\gamma$th quantile of the marginal observations distribution, i.e., $p(y \leq \tau) = \gamma$. This approach leads to a maximum likelihood estimate for the classifier $\hat{\pi}$, which may not properly account for the uncertainty in the classifier's approximation.

Since BORE is based on probabilistic classifiers, instead of regression models as in traditional BO frameworks [1], a natural first question to ask is whether a classifier can guide it to the global optimum of the objective function. The following lemma answers this question and is a basis for our analysis.

**Lemma 1.** *Let $f : \mathcal{X} \to \mathbb{R}$ be a continuous function over a compact space $\mathcal{X}$. Assume that, for any $\mathbf{x} \in \mathcal{X}$, we observe $y = f(\mathbf{x}) + \epsilon$, where $\epsilon$ is i.i.d. noise with a strictly monotonic cumulative distribution function $\Phi_\epsilon : \mathbb{R} \to [0, 1]$. Then, for any $\tau \in \mathbb{R}$, we have:*

$$\operatorname*{argmax}_{\mathbf{x} \in \mathcal{X}} p(y \leq \tau | \mathbf{x}, f) = \operatorname*{argmin}_{\mathbf{x} \in \mathcal{X}} f(\mathbf{x}). \tag{7}$$

---

[2]Note that $p(y \leq s) = \int_{\mathcal{X}} p(y \leq s | \mathbf{x}) p(\mathbf{x}) \, d\mathbf{x}$, where we may assume $p(\mathbf{x})$ uniform.

[3]Bergstra et al. [15] and Tiao et al. [9] also rely on the mild assumption that $p(\mathbf{x}|y) \approx \ell(\mathbf{x})$ for all $y \leq \tau$.

*Proof.* As the observation noise CDF is monotonic, by basic properties of the argmax, we have:

$$\underset{\mathbf{x} \in \mathcal{X}}{\operatorname{argmax}}\, p(y \leq \tau | \mathbf{x}, f) = \underset{\mathbf{x} \in \mathcal{X}}{\operatorname{argmax}}\, \Phi_\epsilon(\tau - f(\mathbf{x})) = \underset{\mathbf{x} \in \mathcal{X}}{\operatorname{argmin}}\, f(\mathbf{x}), \tag{8}$$

which concludes the proof. $\square$

According to this lemma, maximising class probabilities is equivalent to optimising the objective function when the classifier is optimal, i.e., it has perfect knowledge of $f$. This result holds for any given threshold $\tau \in \mathbb{R}$. We only make a mild assumption on the CDF of the observation noise $\Phi_\epsilon$, which is satisfied for any probability distribution with support covering the real line (e.g. Gaussian, Student-T, Cauchy, etc.).[4]

To analyse BORE's optimisation performance, we will aim to bound the algorithm's instant regret:

$$r_t := f(\mathbf{x}_t) - f(\mathbf{x}^*), \quad t \geq 1, \tag{9}$$

and its cumulative version $R_T := \sum_{t=1}^T r_t$. Sub-linear bounds on $R_T$ lead to a no-regret algorithm, since $\lim_{T \to \infty} \frac{R_T}{T} = 0$ and $\min_{t \leq T} r_t \leq \frac{R_T}{T}$.

Assuming that there is an optimal classifier $\pi^* : \mathcal{X} \to [0, 1]$, which is such that $\pi^*(\mathbf{x}) = p(y \leq \tau | \mathbf{x}, f)$, for a given $\tau \in \mathbb{R}$, we can directly relate the classifier probabilities to the objective function $f$ values, since:

$$\pi^*(\mathbf{x}) = p(y \leq \tau | \mathbf{x}, f) = \Phi_\epsilon(\tau - f(\mathbf{x})) \quad \therefore \quad f(\mathbf{x}) = \tau - \Phi_\epsilon^{-1}(\pi^*(\mathbf{x})). \tag{10}$$

The existence of the inverse $\Phi_\epsilon^{-1}$ is ensured by the strict monotonicity assumption on $\Phi_\epsilon$ in Lemma 1. Under this observation, the algorithm's regret at any iteration $t \geq 1$ can be bounded in terms of classifier probabilities:

$$r_t = f(\mathbf{x}_t) - f(\mathbf{x}^*) = \Phi_\epsilon^{-1}(\pi^*(\mathbf{x}^*)) - \Phi_\epsilon^{-1}(\pi^*(\mathbf{x}_t)) \leq L_\epsilon(\pi^*(\mathbf{x}^*) - \pi^*(\mathbf{x}_t)), \tag{11}$$

where $L_\epsilon$ is any Lipschitz constant for $\Phi_\epsilon^{-1}$, which exists since $\mathcal{X}$ is compact. Therefore, we should be able to bound BORE's regret by analysing the approximation error for $\hat{\pi}_t$ at each iteration $t \geq 1$.

Although approximation guarantees for classification algorithms under i.i.d. data settings are well known [18], each observation in BORE depends on the previous ones via the acquisition function. This process is also not necessarily stationary, so that we cannot apply known results for classifiers under stationary processes [19]. In the next section, we consider a particular setting for learning a classifier which allows us to bound the prediction error under BORE's data-generating process.

### 3.1 Probabilistic least-squares classifiers

We consider the case of probabilistic least-squares (PLS) classifiers [20, 21]. In particular, we model a probabilistic classifier $\pi : \mathcal{X} \to [0, 1]$ as an element of a reproducing kernel Hilbert space (RKHS) $\mathcal{H}$ associated with a positive-definite kernel $k : \mathcal{X} \times \mathcal{X} \to \mathbb{R}$. A RKHS is a space of functions equipped with inner product $\langle \cdot, \cdot \rangle_k$ and norm $\|\cdot\|_k := \sqrt{\langle \cdot, \cdot \rangle_k}$ [22]. For the purposes of this analysis, we will also assume that $k(\mathbf{x}, \mathbf{x}) \leq 1$, for all $\mathcal{X}$.[5] This setting allows for both linear and non-parametric models. Gaussian assumptions on the function space would lead us to GP-based PLS classifiers [2], but we are not restricted by Gaussianity in our analysis. If the kernel $k$ is universal, as $\Phi_\epsilon$ is injective, we can also see that the RKHS assumption allows for modelling any continuous function.

For a given $\tau \in \mathbb{R}$, a PLS classifier is obtained by minimising the regularised squared-error loss:

$$\hat{\pi}_t \in \underset{\pi \in \mathcal{H}}{\operatorname{argmin}} \sum_{i=1}^t (z_i - \pi(\mathbf{x}_i))^2 + \lambda \|\pi\|_k^2, \quad t \geq 1, \tag{12}$$

where $\lambda > 0$ is a given regularisation factor and $z_i := \mathbb{I}[y_i \leq \tau] \in \{0, 1\}$. In the RKHS case, the solution to the problem above is available in closed form [23, 16] as:

$$\hat{\pi}_t(\mathbf{x}) = \mathbf{k}_t(\mathbf{x})^\mathsf{T} (\mathbf{K}_t + \lambda \mathbf{I})^{-1} \mathbf{z}_t, \quad \mathbf{x} \in \mathcal{X}, t \geq 1, \tag{13}$$

---

[4]This result could also be easily extended to distributions with bounded support as long as their CDF is monotonic within it. However, we keep the support as $\mathbb{R}$ for simplicity, and the extension is left for future work.

[5]This assumption can always be satisfied by proper scaling.

where $\mathbf{k}_t(\mathbf{x}) := [k(\mathbf{x}, \mathbf{x}_1), \ldots, k(\mathbf{x}, \mathbf{x}_t)]^\mathsf{T} \in \mathbb{R}^t$, $\mathbf{K}_t := [k(\mathbf{x}_i, \mathbf{x}_j)]_{i,j=1}^t \in \mathbb{R}^{t \times t}$ and $\mathbf{z}_t := [z_1, \ldots, z_t]^\mathsf{T} \in \mathbb{R}^t$. This PLS approximation may not yield a valid classifier, since it is possible that $\hat{\pi}_t(\mathbf{x}) \notin [0,1]$ for some $\mathbf{x} \in \mathcal{X}$. However, it allows us to place a confidence interval on the optimal classifier's prediction, as presented in the following theorem, which is based on theoretical results from the online learning literature [24, 25]. Our proofs can be found in the supplement.

**Theorem 1.** *Given $\tau \in \mathbb{R}$, assume $\pi(\mathbf{x}) := \Phi_\epsilon(\tau - f(\mathbf{x}))$ is such that $\pi \in \mathcal{H}$, and $\|\pi\|_k \leq b$. Let $\{\mathbf{x}_t\}_{t=1}^\infty$ be a $\mathcal{X}$-valued discrete-time stochastic process predictable with respect to the filtration $\{\mathfrak{F}_t\}_{t=0}^\infty$. Let $\{z_t\}_{t=1}^\infty$ be a real-valued stochastic process such that $\nu_t := z_t - \pi(\mathbf{x}_t)$ is 1-sub-Gaussian conditionally on $\mathfrak{F}_{t-1}$, for all $t \geq 1$. Then, for any $\delta \in (0,1)$, with probability at least $1 - \delta$, we have that:*

$$\forall \mathbf{x} \in \mathcal{X}, \quad |\pi(\mathbf{x}) - \hat{\pi}_t(\mathbf{x})| \leq \beta_t(\delta)\sigma_t(\mathbf{x}), \quad \forall t \geq 1, \tag{14}$$

*where $\beta_t(\delta) := b + \sqrt{2\lambda^{-1}\log(|\mathbf{I} + \lambda^{-1}\mathbf{K}_t|^{1/2}/\delta)}$, with $|\mathbf{A}|$ denoting the determinant of matrix $\mathbf{A}$, and $\sigma_t^2(\mathbf{x}) := k(\mathbf{x}, \mathbf{x}) - \mathbf{k}_t(\mathbf{x})^\mathsf{T}(\mathbf{K}_t + \lambda\mathbf{I})^{-1}\mathbf{k}_t(\mathbf{x}), \quad \mathbf{x} \in \mathcal{X}, \quad t \geq 1$.*

### 3.2 Regret analysis for BORE

We now consider BORE with a PLS classifier. For this analysis, we will assume an ideal setting where $\tau$ is fixed, possibly corresponding to the true $\gamma$th quantile of the observations distribution. However, our results hold for any choice of $\tau \in \mathbb{R}$ and can therefore be assumed to approximately hold for a varying $\tau$ which is converging to a fixed value. In this setting, the algorithm's choices are: given by:

$$\mathbf{x}_t \in \underset{\mathbf{x} \in \mathcal{X}}{\operatorname{argmax}} \hat{\pi}_{t-1}(\mathbf{x}), \tag{15}$$

where $\hat{\pi}_t$ is the estimator in Equation 13. we can then apply Theorem 1 to the classifier-based regret in Equation 11 to obtain a regret bound. For this result, we will also need the following quantity:

$$\xi_N := \max_{\{\mathbf{x}_i\}_{i=1}^N \subset \mathcal{X}} \frac{1}{2}\log|\mathbf{I} + \lambda^{-1}\mathbf{K}_N|, \quad N \geq 1, \tag{16}$$

where the maximisation is taken over the discrete set of locations $\{\mathbf{x}_i\}_{i=1}^N \subset \mathcal{X}$ and $\mathbf{K}_N := [k(\mathbf{x}_i, \mathbf{x}_j)]_{i,j=1}^N$. This quantity denotes the maximum information gain of a Gaussian process model after $N$ observations. We are now ready to state our theoretical result regarding BORE's regret.

**Theorem 2.** *Under the conditions in Theorem 1, with probability at least $1 - \delta$, $\delta \in (0,1)$, the instant regret of the BORE algorithm with a PLS classifier after $T \geq 1$ iterations is bounded by:*

$$r_t \leq L_\epsilon \beta_{t-1}(\delta)(\sigma_{t-1}(\mathbf{x}_t) + \sigma_{t-1}(\mathbf{x}^*)), \tag{17}$$

*and the cumulative regret by:*

$$R_T \leq L_\epsilon \beta_T(\delta)\left(\sqrt{4(T+2)\xi_T} + \sum_{t=1}^T \sigma_{t-1}(\mathbf{x}^*)\right). \tag{18}$$

As Theorem 2 shows, the regret of the BORE algorithm in the PLS setting is comprised of two components. The first term is related to the regret of a GP-UCB algorithm [see 26, Thr. 3] and its known to grow sub-linearly for a few popular kernels, such as the squared exponential and the Matérn class [3, 27]. The second term, however, reflects the uncertainty of the algorithm around the optimum location $\mathbf{x}^*$. If the algorithm never samples at that location, this second summation might have a mostly linear growth, which will not lead to a vanishing regret. In fact, if we consider Equation 13 and a RKHS with a translation-invariant kernel, we see that, as soon as an observation $z_t = 1$ is collected at a location $\mathbf{x}_t \neq \mathbf{x}^*$, that location will constitute the maximum of the classifier output. Then the algorithm would keep returning to that same location, missing opportunities to sample at $\mathbf{x}^*$.

It is worth noting that Theorem 2 reflects the regret of BORE in an idealistic setting where the algorithm uses the optimal PLS estimator in the function space $\mathcal{H}$. However, if we train a parametric classifier, such as a neural network, via gradient descent, the behaviour will not necessarily be the same, and the algorithm might still achieve a good performance. In the original BORE paper, for instance, a parametric classifier is trained by minimising the binary cross-entropy loss [9] and leads to a successful performance in experiments. Neural network models trained via stochastic gradient

descent are known to provide approximate samples of a posterior distribution [28, 29], instead of an optimal best-fit predictor, which might make BORE behave like Thompson sampling [30] (see discussion in the appendix). Nevertheless, Theorem 2 still shows us that BORE may get stuck into local optima, which is not ideal for BO methods. In the next section, we present an extension of the BORE framework which addresses this shortcoming.

# 4 BORE++: improved uncertainty estimates

As discussed in the previous section, the lack of uncertainty quantification in the estimation of the classifier for the original BORE might lead to sub-optimal performance. To address this shortcoming, we present an approach for uncertainty quantification in the BORE framework which leads to improvements in performance and theoretical optimality guarantees. Our approach is based on using an upper confidence bound (UCB) on the predicted class probabilities as the acquisition function for BORE. Due to its improved uncertainty estimates, we call this approach BORE++.

## 4.1 Class-probability upper confidence bounds

We propose replacing $\hat{\pi}_t$ in Algorithm 1 by an upper confidence bound which is such that:

$$\forall t \geq 1, \quad \pi^*(\mathbf{x}) \leq \pi_{t,\delta}(\mathbf{x}), \quad \forall \mathbf{x} \in \mathcal{X} \tag{19}$$

which with probability greater than $1 - \delta$, given $\delta \in (0, 1)$. Therefore, $\pi_{t,\delta}(\mathbf{x})$ represents an upper quantile over the optimal class probability $\pi^*(\mathbf{x})$. BORE++ selects $\mathbf{x}_t \in \arg\max_{\mathbf{x} \in \mathcal{X}} \pi_{t-1,\delta}(\mathbf{x})$.

To derive an upper confidence bound on a classifier's predictions $\pi(\mathbf{x})$, we can take a few different approaches. For a parametric model $\pi_{\boldsymbol{\theta}}$, a Bayesian model updating the posterior $p(\boldsymbol{\theta}|\mathcal{D}_t)$ leads to a corresponding predictive distribution over $\pi_{\boldsymbol{\theta}}(\mathbf{x})$. This is the case of ensemble models [31], for instance, where we approximate predictions $p(y \leq \tau_t|\mathbf{x}, \mathcal{D}_t) \approx \frac{1}{M} \sum_{i=1}^{M} \pi_{\boldsymbol{\theta}^i}(\mathbf{x})$ with $\boldsymbol{\theta}^i \sim p(\boldsymbol{\theta}|\mathcal{D}_t)$. Instead of using the expected class probability, however, BORE++ uses an (empirical) quantile approximation for $\pi_{t,\delta}$ to ensure Equation 19 holds. Bayesian neural networks [32], random forests [33], dropout methods, etc. [34], also constitute valid approaches for predictive uncertainty estimation. An alternative approach is to place a non-parametric prior over $\pi^*$, such as a Gaussian process model [2], which allows for the modelling of uncertainty directly in the function space where $\pi^*$ lies. In the next section, we present a concrete derivation of BORE++ for the PLS classifier setting which takes the non-parametric perspective and allows us to derive theoretical performance guarantees.

## 4.2 BORE++ with PLS classifiers

In the PLS setting, the result in Theorem 1 gives us a closed-form expression for a classifier upper confidence bound satisfying the condition in Equation 19. Given $\delta \in (0, 1)$, we set:

$$\pi_{t,\delta}(\mathbf{x}) := \min(1, \max(0, \hat{\pi}_t(\mathbf{x}) + \beta_t(\delta)\sigma_t(\mathbf{x}))) \in [0, 1], \quad \mathbf{x} \in \mathcal{X}, \tag{20}$$

where $\sigma_t$ and $\beta_t$ are set according to Theorem 1. We then obtain the following result for BORE++.

**Theorem 3.** *Under the assumptions in Theorem 1, running the BORE++ algorithm with a PLS classifier $\pi_{t,\delta}$ as defined above yields, with probability at least $1 - \delta$, an instant regret bound of:*

$$r_t \leq 2L_\epsilon \beta_t(\delta)\sigma_t(\mathbf{x}), \quad \forall t \geq 1, \tag{21}$$

*and a cumulative regret bound after $T \geq 1$ iterations:*

$$R_T \leq 4L_\epsilon \beta_T(\delta)\sqrt{(T+2)\xi_T} \in \mathcal{O}\left(\sqrt{T}(b\sqrt{\xi_T} + \xi_T)\right). \tag{22}$$

According to Theorem 3, the regret of BORE++ vanishes if the maximum information gain $\xi_T$ grows sub-linearly, since $\lim_{T \to \infty} \frac{R_T}{T} = 0$ and $\min_{t \leq T} r_t \leq \frac{R_T}{T}$. Sub-linear growth is known to be achieved for popular kernels, such as the squared exponential, the Matérn family and linear kernels [3, 27]. This result also tells us that theoretically BORE++ performs no worse than GP-UCB since they share similar regret bounds [3, 26]. However, in practice, the BORE++ framework offers a series of practical advantages over GP-UCB, such as no need for an explicit surrogate model, and a classifier which does not need to be a GP and can therefore be more flexible and scalable to high-dimensional problems and large amounts of data. The connection with GP-UCB, instead, brings us new insights into how the density-ratio BO algorithm can still share some of the well known guarantees of traditional BO methods.

# 5 Batch BORE

This section proposes an extension of the BORE framework which allows for multiple queries to the objective function to be performed in parallel. Although many methods for batch BO have been previously proposed in the literature, we here focus on approaching batch optimisation as an approximate Bayesian inference problem. Instead of having to derive complex heuristics to approximate the utility of a batch of query points, we can view points in a batch as samples from a posterior probability distribution which uses the acquisition function as a likelihood.

## 5.1 BORE batches via approximate inference

Applying an optimisation-as-inference perspective to BORE, we can formulate a batch BO algorithm which does not require an explicit regression model for $f$. The classifier $\hat{\pi}(\mathbf{x}) \approx p(y \leq \tau | \mathbf{x})$ naturally turns out as a likelihood function over query locations $\mathbf{x} \in \mathcal{X}$. Since the search space $\mathcal{X}$ is compact, we can assume a uniform prior distribution $p(\mathbf{x}) \propto 1$. Also note that the normalisation constant in this case is simply $\int_{\mathcal{X}} p(y \leq \tau | \mathbf{x}) p(\mathbf{x}) \, \mathrm{d}\mathbf{x} = p(y \leq \tau) = \gamma$. Our posterior distribution then becomes:

$$\ell(\mathbf{x}) = p(\mathbf{x} | y \leq \tau) = \frac{p(y \leq \tau | \mathbf{x}) p(\mathbf{x})}{p(y \leq \tau)} . \tag{23}$$

Therefore, we formulate a batch version of BORE as an inference problem aiming for:

$$q^* \in \operatorname*{argmin}_{q \in \mathcal{P}} D_{\mathrm{KL}}(q || \ell), \tag{24}$$

where $D_{\mathrm{KL}}(q || \ell)$ denotes the Kullback-Leibler (KL) divergence between $q$ and $\ell$, and $\mathcal{P}$ represents the space of probability distributions over $\mathcal{X}$. Sampling from $\ell$ would allow us to obtain the points of interest in the search space, including the optimum $\mathbf{x}^*$ and other locations where $y \leq \tau$. However, as the true $p(y \leq \tau | \mathbf{x})$ is unknown, we instead formulate a proxy inference problem with respect to a surrogate target distribution $\hat{p}_t$ based on the classifier model. For BORE, we set $\hat{p}_t(\mathbf{x}) \propto \hat{\pi}_{t-1}(\mathbf{x})$, while for BORE++ the setting is $\hat{p}_t(\mathbf{x}) \propto \pi_{t-1,\delta}(\mathbf{x})$. In contrast to $\ell(\mathbf{x}) \propto p(y \leq \tau | \mathbf{x})$, the normalisation constant for the surrogate distributions is unknown, leading us to a proxy problem of minimising $q_t \in \operatorname{argmin}_{q \in \mathcal{P}} D_{\mathrm{KL}}(q || \hat{p}_t)$ at each iteration $t \geq 1$. This variational inference problem above can be efficiently solved via Stein variational gradient descent (SVGD) [35], described next.

## 5.2 Batch sampling via Stein variational gradient descent

In our implementation, we apply SVGD to approximately sample a batch $\mathcal{B}_t := \{\mathbf{x}_{t,i}\}_{i=1}^{M}$ of $M \geq 1$ points from $\hat{p}_t$. Other approximate inference algorithms could also be applied. One of the main advantages of SVGD, however, is that it encourages diversification in the batch, capturing the possible multimodality of $\hat{p}_t$. Given the batch locations, observations can be collected in parallel and then added to the dataset to update the classifier model.

SVGD is an approximate inference algorithm which represents a variational distribution $q$ as a set of particles $\{\mathbf{x}^i\}_{i=1}^{M}$ [35]. The particles are initialised as i.i.d. samples from an arbitrary base distribution and then optimised via a sequence of smooth transformations towards the target distribution, which in our case corresponds to $\hat{p}_t \propto \pi_{t-1,\delta}$. The SVGD steps are given by:

$$\mathbf{x}_t^i \leftarrow \mathbf{x}_t^i + \alpha \boldsymbol{\zeta}_t(\mathbf{x}_t^i), \quad i \in \{1, \ldots, M\}, \tag{25}$$

$$\boldsymbol{\zeta}_t(\mathbf{x}) := \frac{1}{M} \sum_{j=1}^{M} k(\mathbf{x}_t^j, \mathbf{x}) \nabla_{\mathbf{x}_t^j} \log \pi_{t-1,\delta}(\mathbf{x}^j) + \nabla_{\mathbf{x}_t^j} k(\mathbf{x}_t^j, \mathbf{x}), \tag{26}$$

where $k : \mathcal{X} \times \mathcal{X} \to \mathbb{R}$ is a positive-definite kernel, and $\alpha > 0$ is a small step size. Intuitively, the first term in the definition of $\boldsymbol{\zeta}_t$ guides the particles to the modes of $\hat{p}_t$, while the second term encourages diversification by repelling nearby particles. Theoretical convergence guarantees [36, 37] and practical extensions, such as second-order methods [38, 39] and derivative-free approaches [40], have been proposed in the literature. Further details on SVGD can be found in Liu and Wang [35].

## 5.3 Regret bound for Batch BORE++ with PLS classifiers

We follow the derivations in Oliveira et al. [41] to derive a distributional regret bound for batch BORE++ with respect to its target sampling distribution $\ell$, which is presented in the following result.

**Theorem 4.** *Under the same assumptions in Theorem 1, running batch BORE++ with $\pi_{t,\delta}$ set as in Equation 20, we obtain a bound on the instantaneous distributional regret:*

$$\bar{r}_t := \mathbb{E}_{\mathbf{x}\sim\hat{p}_t}[f(\mathbf{x})] - \mathbb{E}_{\mathbf{x}\sim\ell}[f(\mathbf{x})] \leq 2L_\epsilon L_\pi \beta_{t-1}(\delta)\mathbb{E}_{q_t}[\sigma_{t-1}], \quad t \geq 1, \tag{27}$$

*where $L_\pi := \max_{\mathbf{x}\in\mathcal{X}} \frac{1}{\pi(\mathbf{x})}$, and on the cumulative distributional regret at $T \geq 1$:*

$$\bar{R}_T := \sum_{t=1}^{T} \bar{r}_t \leq 4L_\epsilon L_\pi \beta_T(\delta)\sqrt{(T+2)\xi_T} \in \mathcal{O}(\sqrt{T}(b\sqrt{\xi_T} + \sqrt{\xi_T\xi_{MT}})) \tag{28}$$

*both of which hold with probability at least $1 - \delta$.*

As in the case of non-batch BORE++, the distributional regret bounds for the batch algorithm also grow sub-linearly for most popular kernels, leading to an asymptotically vanishing simple regret. Although different, to compare the distributional regret of batch BORE++ with the non-distributional regret bounds for BORE++, we may consider a case where $\tau$ is set to the function minimum $\tau := f(\mathbf{x}^*) = \min_{\mathbf{x}\in\mathcal{X}} f(\mathbf{x})$ and the observation noise is small. In this case, the batch sampling distribution would converge to a Dirac at the optimum, so that $\mathbb{E}_{\mathbf{x}\sim\ell}[f(\mathbf{x})] \approx f(\mathbf{x}^*)$. Compared to the regret of non-batch BORE++ (Theorem 3) after collecting an equivalent number of observations $T' := MT$, the expected regret of the batch version of BORE++ after $T$ iterations is then lower by a factor of $\xi_T/\xi_{MT}$, noting that $\xi_T \leq \xi_{T'} = \xi_{MT}$. Therefore, batch BORE++ should be able to achieve lower regret per runtime than sequential BORE++ with an equivalent number of observations.

## 6    Related work

Since their proposal by Schonlau et al. [42], batch Bayesian optimisation methods have appeared in various forms in the literature. Many methods are based on heuristics derived from estimates given by a Gaussian process regression model [11, 12, 43]. Others are based on Monte Carlo estimates of multi-query acquisition functions [10, 13], optimising points over GP posterior samples [44], solving local optimisation problems [45], or optimising over ensembles of acquisition functions [46]. Despite that, the prevalent approaches to batch BO are still based on a GP regression model, which require prior knowledge about the objective function and do not scale to high-dimensional problems. We instead take a different approach by viewing BO as a density-ratio estimation problem following the BORE framework by Tiao et al. [9]. For batch design, we take an optimisation-as-inference approach [47, 48] by applying Stein variational gradient descent, a non-parametric approximate inference method [35], which has been recently combined with GP-based BO [49, 50]. Our theoretical results, however, are agnostic to the choice of inference algorithm. In contrast to traditional batch BO methods, the inference approach does not require solving inter-dependent optimisation problems for each batch point, as in heuristic-based approaches [43, 11, 12], Monte Carlo integration over the GP posterior [10, 13], nor sampling from it [44]. SVGD allows batch selection to be solved in a vectorised way, which can take advantage of hardware accelerators, such as GPUs.

## 7    Experiments

This section presents experiments assessing the theoretical results and demonstrating the practical performance of batch BORE on a series of global optimisation benchmarks. We compared our methods against GP-based BO baselines in both experiments sets. Additional experimental results, including the sequential setting (Appendix E), a description of the experiments setup (Appendix E), and further discussions on theoretical aspects can be found in the supplementary material.[6]

**Theory assessment.**   We first present simulated experiments assessing the theoretical results in practice, testing BORE and BORE++ in the PLS setting. As a baseline, we compare both methods against GP-UCB. This experiment was run by generating a random base classifier in the RKHS $\mathcal{H}_k$ and then a corresponding objective function via the inverse noise CDF $\Phi_\epsilon^{-1}$. The search space was set as a uniformly-sampled finite subset of the unit interval $\mathcal{X} := [0,1] \subset \mathbb{R}$. We applied the theory-backed settings for BORE++ (Section 2.2) and GP-UCB [25], while BORE employed the optimal PLS classifier (Equation 13).

---

[6]Code will be made available at `https://github.com/rafaol/batch-bore-with-guarantees`

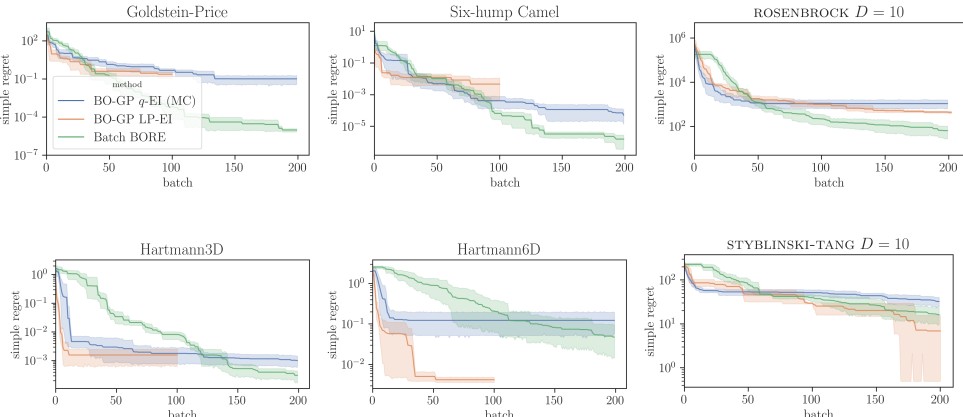

Figure 2: Performance on synthetic benchmarks. Plots show the simple regret, i.e., $\min_{t \leq T} r_t$, per iteration. Results were averaged over 5 trials, and shaded areas indicate the 95% confidence interval.

As the results in Figure 1 show, BORE using an optimal PLS classifier simply gets stuck at a its initial point, resulting in constant regret. BORE++, however, is able to progress in the optimisation problem towards the global optimum, outperforming the GP-UCB baseline.

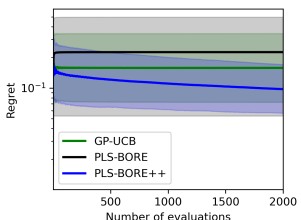

**Global optimisation benchmarks.** We evaluated the proposed SVGD-based batch BORE method in a series of test functions for global optimisation comparing it against other BO baselines. In particular, for our comparisons, we ran the locally penalised EI (LP-EI) method [11] and the Monte Carlo based $q$-EI method [10], which are both based on the EI algorithm, like BORE. Results are presented in Figure 2. All methods ran for $T := 200$ iterations and used of batch size of 10 evaluations per iteration. Additional experimental details are deferred to the supplementary material.

Figure 1: Regret in theory assessment experiment. Results were averaged over 10 trials, and the shaded area indicates the 95% confidence interval.[7]

As Figure 2 shows, batch BORE is able to outperform its baselines on most of the global optimisation benchmarks. We also note that, in some case, due to its complexity the LP-EI method becomes computationally infeasible after 100 iterations, having to be aborted halfway through the optimisation. Batch BORE, however, is able to maintain steady performance throughout its runs.

**Real-data benchmarks.** Lastly, we compared the sequential version of BORE++ against BORE and other baselines, including traditional BO methods, such as GP-UCB and GP-EI [1], the Tree-structured Parzen Estimator (TPE) [15], and random search, on real-data benchmarks. In particular, we assessed the algorithms on some of the same benchmarks present in the original BORE paper [9].

Results are presented in Figure 3. As the plots show, BORE++ presents significantly better performance than BORE in the probabilistic least-squares (PLS) setting (i.e., $\beta_t := 0$), as the theoretical results suggested. In fact, it is possible to note that BORE (PLS) performs comparably to (or at times worse than) random search, indicating that the optimal least-squares classifier by itself is unable to properly capture the epistemic uncertainty. By using a neural network classifier trained via gradient descent and a different loss function (cross-entropy), the original BORE is still able to achieve top performance in most benchmarks. Both BORE versions are only surpassed by traditional GP-based BO on the racing line optimisation problem, as observed in Tiao et al. [9], due to the inherent smoothness the problem, and in the final iterations of the neural architecture search problem by GP-EI. Interestingly, even though restricted to the kernel-based PLS setting, we observe that BORE++ is able to surpass the original BORE in the neural network hyper-parameter tuning problems (SLICE

---

[7]Linear interpolation is applied to obtain the plotted confidence intervals when the number of trials is small.

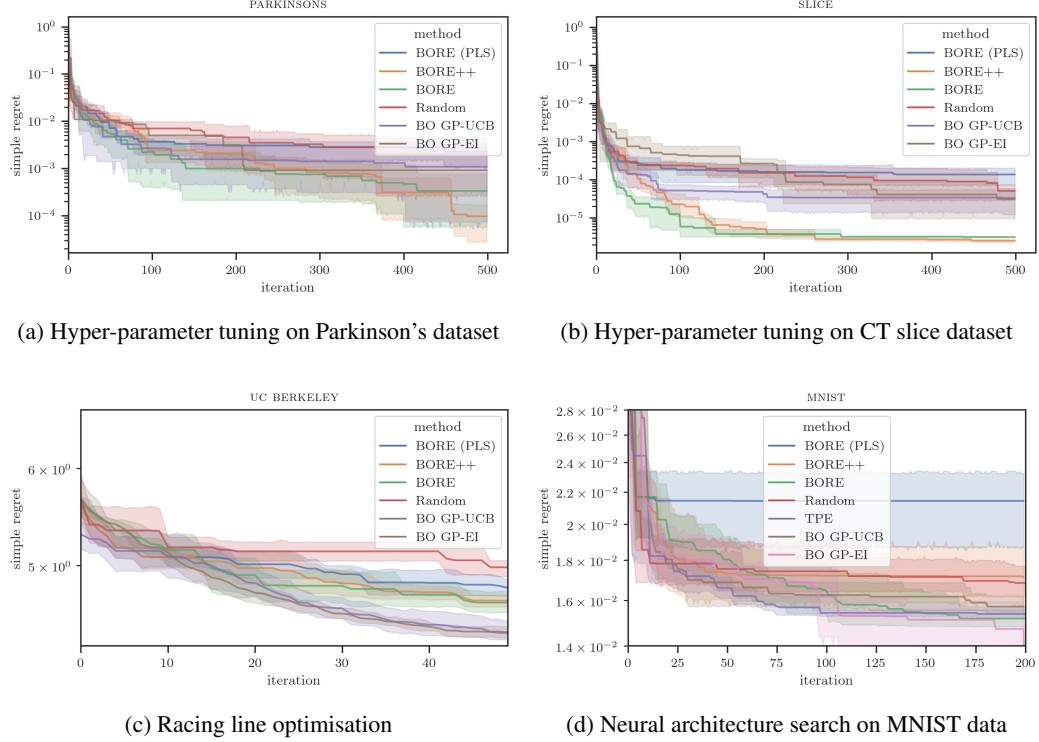

(a) Hyper-parameter tuning on Parkinson's dataset

(b) Hyper-parameter tuning on CT slice dataset

(c) Racing line optimisation

(d) Neural architecture search on MNIST data

Figure 3: Experimental results on real-data benchmarks. The plots show each algorithm's simple regret averaged across multiple runs. The shaded areas correspond to the 95% confidence intervals.

and PARKINSONS), while maintaining similar performance in other tasks. These results confirm that improved uncertainty estimates can lead to practical performance gains.

## 8 Conclusion

This paper presented an extension of the BORE framework to the batch setting alongside the theoretical analysis of the proposed extension and an improvement over the original BORE. Theoretical results in terms of regret bounds and experiments show that BORE methods are able to maintain performance guarantees while outperforming traditional BO baselines. The main purpose of this work, however, was to establish the theoretical foundations for the analysis and derivation of new algorithmic frameworks for Bayesian optimisation via density-ratio estimation, equipping BO with new tools based on probabilistic classification, instead of regression models.

As future work, we plan to investigate the theoretical properties of BORE under different loss functions and analyse other batch design strategies, e.g., sampling methods for combinatorial domains. The theoretical contributions of this work can also be extended to other versions of BORE, such as its recent multi-objective version [51], and provide insights into other likelihood-free BO methods [52]. Finally, we consider integrating BORE++ with other probabilistic classification models equipped with predictive uncertainty estimates, such as neural network ensembles [53], random forests [54], and Bayesian generalised linear models, which should lead to improvements in scalability and additional performance gains.

## Acknowledgments and Disclosure of Funding

Rafael Oliveira was supported by the Medical Research Future Fund Applied Artificial Intelligence in Health Care grant (MRFAI000097) by the Australian Department of Health and Aged Care.

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
