# Batch Bayesian optimisation via density-ratio estimation with guarantees

**Rafael Oliveira**[1,2*]
rafael.oliveira@sydney.edu.au

**Louis C. Tiao**[3]
louis.tiao@sydney.edu.au

**Fabio Ramos**[3,4]
fabio.ramos@sydney.edu.au

[1]Brain and Mind Centre, the University of Sydney, Australia
[2]ARC Training Centre in Data Analytics for Resources and Environments, Australia
[3]School of Computer Science, the University of Sydney, Australia
[4]NVIDIA, USA

## Appendix

This appendix complements the main paper with proofs, experiment details and additional experiments and discussions. Appendix A presents full proofs for the main theoretical results in the paper. In Appendix B, we discuss an approach to derive alternative regret bounds for BORE under a Thompson sampling perspective. We discuss the theoretical analysis of BORE with its non-constant approximation for the observations quantile $\tau$ in Appendix C. In Appendix D, we present further details on the experiments setup. Finally, Appendix E presents an additional experiment assessing dimensionality effects.

## A  Proofs

This section presents proofs for the main theoretical results in the paper. We start with a few auxiliary results from the GP-UCB literature [1, 2], following up with the proofs for the main theorems.

### A.1  Auxiliary results

**Lemma A.1** (Srinivas et al. [1, Lemma 5.3]). *The information gain for a sequence of $N \geq 1$ observations $\{\mathbf{x}_i, z_i\}_{i=1}^N$, where $z_i = f(\mathbf{x}_i) + \nu_i$, $\nu_i \sim \mathcal{N}(0, \lambda)$, can be expressed in terms of the predictive variances. Namely, if $f \sim \mathcal{GP}(m, k)$, then the information gain provided by the observations is such that:*

$$I(\mathbf{z}_N, \mathbf{f}_N | \mathcal{X}_N) = \frac{1}{2} \sum_{i=1}^N \log(1 + \lambda^{-1} \sigma_{i-1}^2(\mathbf{x}_i)), \tag{A.1}$$

*where $\mathbf{f}_N := [f(\mathbf{x}_i)]_{i=1}^N$ and $\mathcal{X}_N := \{\mathbf{x}_i\}_{i=1}^N \subset \mathcal{X}$.*

**Lemma A.2** (Chowdhury and Gopalan [2, Lemma 4]). *Following the setting of Lemma A.1, the sum of predictive standard deviations at a sequence of $N$ points is bounded in terms of the maximum information gain:*

$$\sum_{i=1}^N \sigma_{i-1}(\mathbf{x}_i) \leq \sqrt{4(N+2)\xi_N}. \tag{A.2}$$

---

*Corresponding author.

36th Conference on Neural Information Processing Systems (NeurIPS 2022).

**Lemma A.3.** *Let $\mathcal{A} \subset \mathcal{X}$ be a finite set of points where a function $f \sim \mathcal{GP}(m, k)$ was evaluated, so that the GP posterior covariance function and the corresponding variance are given by:*

$$k_\mathcal{A}(\mathbf{x}, \mathbf{x}') := k(\mathbf{x}, \mathbf{x}') - k(\mathbf{x}, \mathcal{A})^\mathsf{T}(\mathbf{K}(\mathcal{A}) + \eta\mathbf{I})^{-1}k(\mathcal{A}, \mathbf{x}') \tag{A.3}$$

$$\sigma_\mathcal{A}^2(\mathbf{x}) := k_\mathcal{A}(\mathbf{x}, \mathbf{x}), \quad \mathbf{x}, \mathbf{x}' \in \mathcal{X}, \tag{A.4}$$

*where $k(\mathbf{x}, \mathcal{A}) := [k(\mathbf{x}, \mathbf{a})]_{\mathbf{a} \in \mathcal{A}}$ and $\mathbf{K}(\mathcal{A}) := [k(\mathbf{a}, \mathbf{a}')]_{\mathbf{a}, \mathbf{a}' \in \mathcal{A}}$. Then, for any given set $\mathcal{B} \supset \mathcal{A}$ of evaluations of $f$, we have:*

$$\sigma_\mathcal{B}^2(\mathbf{x}) \leq \sigma_\mathcal{A}^2(\mathbf{x}), \quad \forall \mathbf{x} \in \mathcal{X}. \tag{A.5}$$

*Proof.* The result follows by observing that the GP posterior given observations at $\mathcal{A}$ is a prior for the GP with the new observations at the complement $\mathcal{C} := \mathcal{B} \setminus \mathcal{A}$. Then we obtain, for all $\mathbf{x} \in \mathcal{X}$:

$$\begin{aligned}
\sigma_\mathcal{B}^2(\mathbf{x}) &:= k(\mathbf{x}, \mathbf{x}) - k(\mathbf{x}, \mathcal{B})(\mathbf{K}(\mathcal{B}) + \eta\mathbf{I})^{-1}k(\mathcal{B}, \mathbf{x}) \\
&= \sigma_\mathcal{A}^2(\mathbf{x}) - k_\mathcal{A}(\mathbf{x}, \mathcal{C})(\mathbf{K}_\mathcal{A}(\mathcal{C}) + \eta\mathbf{I})^{-1}k_\mathcal{A}(\mathcal{C}, \mathbf{x}) \\
&\leq \sigma_\mathcal{A}^2(\mathbf{x}),
\end{aligned} \tag{A.6}$$

since $k_\mathcal{A}(\mathbf{x}, \mathcal{C})(\mathbf{K}_\mathcal{A}(\mathcal{C}) + \eta\mathbf{I})^{-1}k_\mathcal{A}(\mathcal{C}, \mathbf{x})$ is non-negative. $\qquad\square$

## A.2 Proof of Theorem 1

*Proof of Theorem 1.* The proof follows by a simple application of Durand et al. [3, Thm. 1] on GP-UCB to our settings, as $\pi \in \mathcal{H}$ and the stochastic process defining the query locations $\mathbf{x}_t$ and observation noise $\nu_t := z_t - \pi(\mathbf{x}_t)$ satisfies their assumptions of sub-Gaussianity. In particular, $\nu_t$ is $\sigma_\nu$-sub-Gaussian with $\sigma_\nu \leq 1$, since $|\nu_t| \leq 1$, for all $t \geq 1$ [4]. $\qquad\square$

## A.3 Proof of Theorem 2

To prove Theorem 2, we will follow the procedure of GP-UCB proofs [1, 2] by bounding the approximation error $|\pi(\mathbf{x}) - \hat{\pi}_t(\mathbf{x})|$ via a confidence bound (Theorem 1) and then applying it to the instant regret. From the instant regret to the cumulative regret, the bounds are extended by means of the maximum information gain $\xi_T$ introduced in the main text. One of the differences with our proof, however, is that BORE with a PLS classifier is not following the optimal UCB policy, but instead a pure-exploitation approach by following the maximum of the mean estimator $\hat{\pi}_t$, which does not account for uncertainty.

*Proof of Theorem 2.* Recalling the classifier-based bound in Section 4 and that for any $\tau \in \mathbb{R}$ the result in Lemma 1 holds, we have:

$$\begin{aligned}
r_t &= f(\mathbf{x}_t) - f(\mathbf{x}^*) \\
&\leq L_\epsilon(\pi(\mathbf{x}^*) - \pi(\mathbf{x}_t))
\end{aligned} \tag{A.7}$$

According to Theorem 1, working with the confidence bounds on $\pi(\mathbf{x})$, we then have that the instant regret is bounded with probability at least $1 - \delta$ by:

$$\begin{aligned}
\forall t \geq 1, \quad r_t &\leq L_\epsilon(\hat{\pi}_{t-1}(\mathbf{x}^*) + \beta_{t-1}(\delta)\sigma_{t-1}(\mathbf{x}^*) - \pi(\mathbf{x}_t)) \\
&\leq L_\epsilon(\hat{\pi}_{t-1}(\mathbf{x}^*) + \beta_{t-1}(\delta)\sigma_{t-1}(\mathbf{x}^*) - \hat{\pi}_{t-1}(\mathbf{x}_t) + \beta_{t-1}(\delta)\sigma_{t-1}(\mathbf{x}_t)) \\
&\leq L_\epsilon\beta_{t-1}(\delta)(\sigma_{t-1}(\mathbf{x}^*) + \sigma_{t-1}(\mathbf{x}_t)),
\end{aligned} \tag{A.8}$$

since $\hat{\pi}_{t-1}(\mathbf{x}^*) \leq \max_{\mathbf{x} \in \mathcal{X}} \hat{\pi}_{t-1}(\mathbf{x}) = \hat{\pi}_{t-1}(\mathbf{x}_t)$. Now we can apply Lemma A.2, yielding with probability at least $1 - \delta$:

$$\begin{aligned}
R_T &:= \sum_{t=1}^T r_t \leq L_\epsilon\beta_T(\delta)\sum_{t=1}^T (\sigma_{t-1}(\mathbf{x}_t) + \sigma_{t-1}(\mathbf{x}^*)) \\
&\leq L_\epsilon\beta_T(\delta)\left(\sqrt{4(T+2)\xi_T} + \sum_{t=1}^T \sigma_{t-1}(\mathbf{x}^*)\right)
\end{aligned} \tag{A.9}$$

since $\beta_t(\delta) \leq \beta_{t+1}(\delta)$ for all $t \geq 1$. This concludes the proof. $\qquad\square$

## A.4 Proof of Theorem 3

Again, we will be following standard GP-UCB proofs for this result using the bound in Theorem 1.

*Proof of Theorem 3.* Extending the bound in [Equation A.7](#) with Theorem 1, we have with probability at least $1 - \delta$:

$$
\begin{aligned}
\forall t \geq 1, \quad r_t &\leq L_\epsilon(\hat{\pi}_{t-1}(\mathbf{x}^*) + \beta_{t-1}(\delta)\sigma_{t-1}(\mathbf{x}^*) - \pi^*_{t-1}(\mathbf{x}_t)) \\
&\leq L_\epsilon(\hat{\pi}_{t-1}(\mathbf{x}^*) + \beta_{t-1}(\delta)\sigma_{t-1}(\mathbf{x}^*) - \hat{\pi}_{t-1}(\mathbf{x}_t) + \beta_{t-1}(\delta)\sigma_{t-1}(\mathbf{x}_t)) \qquad \text{(A.10)} \\
&\leq 2L_\epsilon\beta_{t-1}(\delta)\sigma_{t-1}(\mathbf{x}_t),
\end{aligned}
$$

since $\hat{\pi}_{t-1}(\mathbf{x}^*) + \beta_{t-1}(\delta)\sigma_{t-1}(\mathbf{x}^*) \leq \max_{\mathbf{x}\in\mathcal{X}} \hat{\pi}_{t-1}(\mathbf{x}) + \beta_{t-1}(\delta)\sigma_{t-1}(\mathbf{x}) = \hat{\pi}_{t-1}(\mathbf{x}_t) + \beta_{t-1}(\delta)\sigma_{t-1}(\mathbf{x}_t))$. Turning our attention to the cumulative regret, with the same probability, we have:

$$
\begin{aligned}
R_T := \sum_{t=1}^{T} r_t &\leq 2L_\epsilon\beta_T(\delta)\sum_{t=1}^{T}\sigma_{t-1}(\mathbf{x}_t) \\
&\leq 4L_\epsilon\beta_T(\delta)\sqrt{(T+2)\xi_T},
\end{aligned} \qquad \text{(A.11)}
$$

which concludes the proof. $\qquad\square$

## A.5 Proof of Theorem 4

*Proof.* Starting with the regret definition, we can define a bound in terms of the discrepancy between the two sampling distributions:

$$
\begin{aligned}
r_t &:= \mathbb{E}_{\mathbf{x}\sim\hat{p}_t}[f(\mathbf{x})] - \mathbb{E}_{\mathbf{x}\sim\ell}[f(\mathbf{x})] \\
&\leq L_\epsilon\left(\mathbb{E}_{\mathbf{x}\sim\ell}[\pi(\mathbf{x})] - \mathbb{E}_{\mathbf{x}\sim\hat{p}_t}[\pi(\mathbf{x})]\right) \\
&\leq L_\epsilon\|\pi\|_\infty\int_{\mathcal{X}}|\ell(\mathbf{x}) - q_{t-1}(\mathbf{x})|\,\mathrm{d}\mathbf{x} \qquad \text{(A.12)} \\
&\leq L_\epsilon\|\pi\|_\infty\sqrt{\frac{1}{2}D_{\mathrm{KL}}(q_{t-1}||\ell)}, \qquad \forall t \geq 1,
\end{aligned}
$$

where the last line is due to Pinsker's inequality [4] applied to the total variation distance between $\hat{p}_t$ and $\ell$ (third line).

Tp obtain a bound on $D_{\mathrm{KL}}(\hat{p}_t||\ell)$, starting from the definition of the terms, with probability at least $1 - \delta$, we have that:

$$
\begin{aligned}
\forall t \geq 0, \quad D_{\mathrm{KL}}(\hat{p}_t||\ell) &= \mathbb{E}_{\mathbf{x}\sim\hat{p}_t}[\log\hat{p}_t(\mathbf{x}) - \log\ell(\mathbf{x})] \\
&= \mathbb{E}_{\mathbf{x}\sim\hat{p}_t}[\log(\hat{\pi}_t(\mathbf{x}) + \beta_t(\delta)\sigma_t(\mathbf{x})) - \log\pi(\mathbf{x}) + \log\eta_\pi - \log\gamma] \qquad \text{(A.13)} \\
&\leq \mathbb{E}_{\mathbf{x}\sim\hat{p}_t}[\log(\hat{\pi}_t(\mathbf{x}) + \beta_t(\delta)\sigma_t(\mathbf{x})) - \log\pi(\mathbf{x})],
\end{aligned}
$$

which follows from $\eta_t := \int_{\mathcal{X}}(\hat{\pi}_t(\mathbf{x}) + \beta_t(\delta)\sigma_t(\mathbf{x}))p(\mathbf{x})\,\mathrm{d}\mathbf{x} \geq \int_{\mathcal{X}}\pi(\mathbf{x})p(\mathbf{x})\,\mathrm{d}\mathbf{x} =: \gamma$. Now, by the mean value theorem [5], for all $t \geq 0$, we have that the following holds with the same probability:

$$
\begin{aligned}
|\log(\hat{\pi}_t(\mathbf{x}) + \beta_t(\delta)\sigma_t(\mathbf{x})) - \log\pi(\mathbf{x})| &\leq L_\pi|\hat{\pi}_t(\mathbf{x}) + \beta_t(\delta)\sigma_t(\mathbf{x}) - \pi(\mathbf{x})| \\
&\leq 2L_\pi\beta_t(\delta)\sigma_t(\mathbf{x}), \quad \forall\mathbf{x}\in\mathcal{X},
\end{aligned} \qquad \text{(A.14)}
$$

since $\frac{\mathrm{d}\log(s)}{\mathrm{d}s} < L_\pi < \infty$ for all $s > \min_{\mathbf{x}\in\mathcal{X}}\pi(\mathbf{x}) > 0$, and $|\hat{\pi}_t(\mathbf{x}) - \pi(\mathbf{x})| \leq \beta_t(\delta)\sigma_t(\mathbf{x})$ by Theorem 1. The first result in the theorem then follows.

For the second part of the result, we first note that:

$$
\forall T \geq 1, \quad \min_{t\leq T} D_{\mathrm{KL}}(\hat{p}_t||\ell) \leq \frac{1}{T}\sum_{t=1}^{T} D_{\mathrm{KL}}(\hat{p}_{t-1}||\ell) \qquad \text{(A.15)}
$$

Following the previous derivations, it holds with probability at least $1 - \delta$ that:

$$\sum_{t=1}^{T} D_{\mathrm{KL}}(\hat{p}_t||\ell) \leq 2L_\pi \sum_{t=1}^{T} \beta_{t-1}(\delta)\mathbb{E}_{\tilde{\mathbf{x}}_t \sim \hat{p}_t}[\sigma_{t-1}(\tilde{\mathbf{x}}_t)]$$

$$\leq 2L_\pi \beta_T(\delta) \sum_{t=1}^{T} \mathbb{E}_{\tilde{\mathbf{x}}_t \sim q_t}[\sigma_{t-1}(\tilde{\mathbf{x}}_t)] \tag{A.16}$$

$$\leq 2L_\pi \beta_T(\delta)\mathbb{E}_{\tilde{\mathbf{x}}_1 \sim q_1, \ldots, \tilde{\mathbf{x}}_T \sim q_T}\left[\sum_{t=1}^{T} \sigma_{t-1}(\tilde{\mathbf{x}}_t)\right],$$

since $\beta_T \geq \beta_t$, for all $t \leq T$, and expectations are linear operations. Considering the predictive variances above, recall that, at each iteration $t \geq 1$, the algorithm selects a batch of i.i.d. points $\mathcal{B}_t := \{\mathbf{x}_{t,i}\}_{i=1}^{M}$, sampled from $\hat{p}_t$, where to evaluate the objective function $f$. The predictive variance $\sigma_{t-1}^2$ is conditioned on all previous observations, which are grouped by batches. We can then decompose, for any $t \geq 1$:

$$\sigma_t^2(\mathbf{x}) = \sigma_{t-1}^2(\mathbf{x}) - k_{t-1}(\mathbf{x}, \mathcal{B}_t)(\mathbf{K}_{t-1}(\mathcal{B}_t) + \eta\mathbf{I})^{-1}k_{t-1}(\mathcal{B}_t, \mathbf{x}), \tag{A.17}$$

where we use the notation introduced in Lemma A.3, and:

$$k_t(\mathbf{x}, \mathbf{x}') = k_{t-1}(\mathbf{x}, \mathbf{x}') - k_{t-1}(\mathbf{x}, \mathcal{B}_t)(\mathbf{K}_{t-1}(\mathcal{B}_t) + \eta\mathbf{I})^{-1}k_{t-1}(\mathcal{B}_t, \mathbf{x}'), \quad t \geq 1, \tag{A.18}$$

$$k_0(\mathbf{x}, \mathbf{x}') := k(\mathbf{x}, \mathbf{x}'). \tag{A.19}$$

Therefore, the predictive variance of the batched algorithm is not the same as the predictive variance of a sequential algorithm, and we cannot direcly apply Lemma A.2 to bound the last term in Equation A.16.

Lemma A.3 tells us that the predictive variance given a set of observations is less than the predictive variance given a subset of observations. Selecting only the first point from within each batch and applying Lemma A.3, we get, for $t \geq 1$:

$$\sigma_t^2(\mathbf{x}) \leq s_t^2(\mathbf{x}) := k(\mathbf{x}, \mathbf{x}) - k(\mathbf{x}, \mathcal{X}_t)(\mathbf{K}(\mathcal{X}_t) + \eta\mathbf{I})^{-1}k(\mathcal{X}_t, \mathbf{x}), \tag{A.20}$$

where $\mathcal{X}_t := \{\mathbf{x}_{i,1}\}_{i=1}^{t}$, with $\mathbf{x}_{i,1} \in \mathcal{B}_i$, $i \in \{1, \ldots, t\}$. Note that the right-hand side of the equation above is simply the non-batched GP predictive variance. Furthermore, sample points within a batch are i.i.d., so that $\mathbf{x}_{t,1} \sim q_t$ and $\tilde{\mathbf{x}}_t \sim q_t$ are identically distributed. We can now apply Lemma A.2, yielding:

$$\mathbb{E}_{\tilde{\mathbf{x}}_1 \sim q_1, \ldots, \tilde{\mathbf{x}}_T \sim q_T}\left[\sum_{t=1}^{T} \sigma_{t-1}(\tilde{\mathbf{x}}_t)\right] \leq \mathbb{E}_{\tilde{\mathbf{x}}_1 \sim q_1, \ldots, \tilde{\mathbf{x}}_T \sim q_T}\left[\sum_{t=1}^{T} s_{t-1}(\tilde{\mathbf{x}}_t)\right] \leq 2\sqrt{(T+2)\xi_T}. \tag{A.21}$$

Combining this result with Equation A.16, we obtain:

$$\sum_{t=1}^{T} D_{\mathrm{KL}}(\hat{p}_t||\ell) \leq 4L_\pi \beta_T(\delta)\sqrt{(T+2)\xi_T} \in \mathcal{O}(\beta_T(\delta)\sqrt{T\xi_T}). \tag{A.22}$$

Lastly, from the definition of $\beta_t(\delta)$, we have:

$$\beta_T(\delta) := b + \sqrt{2\lambda^{-1}\log(|\mathbf{I} + \lambda^{-1}\mathbf{K}_{\mathcal{D}_T}|^{1/2}/\delta)}, \tag{A.23}$$

where:

$$\log(|\mathbf{I} + \lambda^{-1}\mathbf{K}_{\mathcal{D}_T}|^{1/2}) = I(\mathbf{z}_{N_T}, \mathbf{h}_{N_T}) \leq \xi_{N_T} = \xi_{MT}, \tag{A.24}$$

for $h \sim \mathcal{GP}(m, k)$. Therefore, the cumulative sum of divergences is such that:

$$\sum_{t=1}^{T} D_{\mathrm{KL}}(\hat{p}_t||\ell) \in \mathcal{O}(\sqrt{T}(b\sqrt{\xi_T} + \sqrt{\xi_T\xi_{MT}})). \tag{A.25}$$

which concludes the proof. □

## B   Bayesian regret bounds for BORE as Thompson sampling

Although in our main results we considered BORE using an optimal classifier according to a least-squares loss, we may instead consider that, in practice, the trained classifier might be sub-optimal due to training via gradient descent. In particular, in the case of stochastic gradient descent, Mandt et al. [6] showed that parameters learnt this way can be seen as approximate samples of a Bayesian posterior distribution. This is, therefore, the case of Thompson (or posterior) sampling [7]. If we consider that the posterior over the model's function space is Gaussian, e.g., as in the case of infinitely-wide deep neural networks [8, 9], we may instead analyse the original BORE as a GP-based Thompson sampling algorithm. We can then apply theoretical results from Russo and Van Roy [7] to use general GP-UCB approximation guarantees [1, 10] to bound BORE'S Bayesian regret. Note, however, that this is a different type of analysis compared to the one presented in this paper, which considered a frequentist setting where the objective function is fixed, but unknown.

## C   Analysis with a non-constant quantile approximation

Our main theoretical results so far relied upon the quantile $\tau$ being fixed throughout all iterations $t \in \{1, \ldots T\}$, though in practice we have to approximate the quantile based on the empirical observations distribution up to time $t \geq 1$. In this section, we discuss the plausibility of the theoretical results under this practical scenario.

The main impact of a time-varying quantile $\tau_t$, and the corresponding classifier $\pi_t(\mathbf{x}) := p(y \leq \tau_t | \mathbf{x})$, in theoretical results is in the UCB approximation error (Theorem 1). This result depends on the observation noise $\nu_{t,i} := z_{t,i} - \pi_t(\mathbf{x}_i)$ as perceived by a GP model with observations $z_{t,i} := \mathbb{I}[y_i \leq \tau_t]$, $i \in \{1, \ldots, t\}$, to be sub-Gaussian when conditioned on the history. Hence, a few challenges originate from there. Firstly, the past observations in the vector $\mathbf{z}_t := [z_{t,i}]_{i=1}^t$ are changing across iterations, due to the update in $\tau_t$. Secondly, the latent function $\pi_t$ is stochastic, as the quantile $\tau_t$ depends on the current set of observations $\mathbf{y}_t$. Lastly, it is not very clear how to define a filtration for the resulting stochastic process such that the GP noise $\nu_{t,i}$ is sub-Gaussian. Nevertheless, as the number of observations increases, $\tau$ converges to a fixed value, making our asymptotic results valid.

## D   Experiment details

This section presents details of our experiments setup. We used PyTorch [11] for our implementation of batch BORE and BORE++, which we plan to make publicly available in the future.

### D.1   Theory assessment

For this experiment, we generated a random classifier as an element of the RKHS $\mathcal{H}$ defined by the kernel $k$ as:

$$\pi^* := \sum_{i=1}^F \alpha_i k(\cdot, \mathbf{x}_i^*) \in \mathcal{H}, \tag{D.1}$$

where $\{\mathbf{x}_i^*\}_{i=1}^F$ and the weights $\{\alpha_i\}_{i=1}^F$ were i.i.d. sampled from a unit uniform distribution $\mathcal{U}(0, 1)$, with $F := 5$. The norm of $\pi^*$ is given by:

$$\|\pi^*\|_k = \sqrt{\boldsymbol{\alpha}_F^\mathsf{T} \mathbf{K}_F \boldsymbol{\alpha}_F}, \tag{D.2}$$

where $\mathbf{K} := [\mathbf{x}_i^*, \mathbf{x}_j^*]_{i,j=1}^F \in \mathbb{R}^{F \times F}$ and $\boldsymbol{\alpha}_F := [\alpha_1, \ldots, \alpha_F]^\mathsf{T} \in \mathbb{R}^F$. To ensure $\pi^*(\mathbf{x}) \leq 1$, we normalised the weights according to the classifier norm, i.e., $\boldsymbol{\alpha} := \frac{1}{\|\pi^*\|} \boldsymbol{\alpha}$, so that $\|\pi^*\| = 1$, and consequently $\pi^*(\mathbf{x}) \leq k(\mathbf{x}, \mathbf{x}) \|\pi^*\| = \|\pi^*\| = 1$, for all $\mathbf{x} \in \mathcal{X}$. The kernel was set as the squared exponential (RBF) with a length-scale of $0.1$.

Given a threshold $\tau \in \mathbb{R}$, the objective function corresponding to $\pi^*$ satisfies:

$$f(\mathbf{x}) := \tau - \Phi_\epsilon^{-1}(\pi^*(\mathbf{x})), \quad \forall \mathbf{x} \in \mathcal{X}. \tag{D.3}$$

For this experiment, we set $\tau := 0$. Each trial had a different objective function generated for it. An example of classifier and objective function pair is presented in Figure 1b (main paper).

| Parameter | Setting |
|:---:|:---:|
| $\lambda$ | 0.025 |
| $\delta$ | 0.1 |
| $\tau$ | 0 |

Table D.1: Settings for BORE++ in the theory assessment experiment.

| Parameter | Setting |
|:---:|:---:|
| $\lambda$ | 0.01 |
| $\delta$ | 0.1 |
| $\sigma_\epsilon^2$ | 0.01 |

Table D.2: Settings for GP-UCB in the theory assessment experiment.

Observations were composed as function evaluations corrupted by zero-mean Gaussian noise with variance $\sigma_\epsilon^2 := 0.01$.

The search space was configured as a finite set $\mathcal{X} := \{\mathbf{x}_i\}_{i=1}^{N_{\mathcal{X}}} \subset [0,1]$ by sampling $N_{\mathcal{X}}$ points from a unit uniform distribution. The number of points in the search space was set as $N_{\mathcal{X}} := 100$. As the search space is finite, we also know $\gamma := p(y \leq \tau) = \int_{\mathcal{X}} \pi(\mathbf{x}) p(\mathbf{x}) \, d\mathbf{x} = \frac{1}{N_{\mathcal{X}}} \sum_{\mathbf{x} \in \mathcal{X}} \pi^*(\mathbf{x})$.

Regarding algorithm hyper-parameters, although any upper bound $b \geq \|\pi^*\|$ would work for setting up $\beta_t$, BORE++ was configured with the RKHS norm $\pi^*$ as the first term in the setting for $\beta_t$ (see Theorem 1). To configure GP-UCB according to its theoretical settings [3, Thm. 1], we computed the RKHS norm of the resulting $f$ in the RKHS. We can compute the norm of $f$ as an element of $\mathcal{H}$ by solving the following constrained optimisation problem:

$$\|f\|_k = \min_{h \in \mathcal{H}} \|h\|_k, \quad \text{s.t.} \quad h(\mathbf{x}) = f(\mathbf{x}), \quad \forall \mathbf{x} \in \mathcal{X}. \tag{D.4}$$

As the search space is finite, the solution to this problem is available in closed form as:

$$\|f\|_k = \sqrt{\mathbf{f}_{\mathcal{X}}^{\mathsf{T}} \mathbf{K}_{\mathcal{X}}^{-1} \mathbf{f}_{\mathcal{X}}}, \tag{D.5}$$

where $\mathbf{f}_{\mathcal{X}} := [f(\mathbf{x})]_{\mathbf{x} \in \mathcal{X}} \in \mathbb{R}^{N_{\mathcal{X}}}$, and $\mathbf{K}_{\mathcal{X}} := [k(\mathbf{x}, \mathbf{x}')]_{\mathbf{x}, \mathbf{x}' \in \mathcal{X}}$. We set $\delta := 0.1$. For both BORE++ and GP-UCB, the information gain was recomputed at each iteration. Lastly, the regularisation factor $\lambda$ was set as $\lambda := \sigma_\epsilon^2$ for GP-UCB and as $\lambda := 0.025$ for BORE++, which was found by grid search. In summary, for this experiment, the settings for BORE++ can be found in Table D.1 and, for GP-UCB, in Table D.2.

### D.2 Global optimisation benchmarks

For each problem, all methods used 10 initial points uniformly sampled from the search space. As performance indicator, we measured the simple regret:

$$r_t^* := \min_{i \leq t} r_i = \min_{i \leq t} f(\mathbf{x}_i) - f(\mathbf{x}^*), \quad t \geq 1, \tag{D.6}$$

as the global minimum of each of the considered benchmark functions is known. All objective function evaluations were provided free of noise to the algorithms.

Batch BORE was run with a percentile $\gamma := 0.25$, which was applied to estimate the empirical quantile $\tau$ at every iteration $t \in \{1, \ldots, T\}$. The method's classifier model was composed of a multilayer perceptron neural network model with 2 hidden layers of 32 units each, which was trained to minimise the binary cross-entropy loss. The activation function was set as the rectified linear unit (ReLU) with exception for the Hartmann 3D and the Six-hump Camel problem, which were run with an exponential linear unit (ELU), instead. Training for the neural networks was performed via stochastic gradient descent, whose settings are presented in Table D.3. SVGD was run applying Adadelta to configure its steps according to the settings in Table D.4. The SVGD kernel was set as the squared exponential (RBF) using the median trick to adjust its lengthscale [12].

LP-EI [13] was run using L-BFGS [14] to optimise its acquisition function. The optimisation settings were kept as the default for GPyOpt [15].

| Parameter | Setting |
|-----------|---------|
| Optimiser | Adam |
| Batch size | 64 |
| Steps | 100* |

Table D.3: Stochastic gradient descent training settings for batch BORE. (*) For the Six-hump Camel problem, we applied 200 steps.

| Parameter | Setting |
|-----------|---------|
| Step size | 0.001 |
| Decay rate | 0.9 |
| Number of steps | 1000* |

Table D.4: SVGD settings for batch BORE. (*) For the Hartmann 3D problem, we used 500 steps.

The $q$-EI method [16] was run using the BoTorch implementation [17]. The acquisition function was optimised via multi-start optimisation with L-BFGS [14] using 10 random restarts. Monte Carlo integration for $q$-EI used 256 samples.

## D.3 Comparisons on real-data benchmarks

We here present experiments comparing the sequential version of BORE++ against BORE and other baselines, including traditional BO methods, such as GP-UCB and GP-EI [18], the Tree-structured Parzen Estimator (TPE) [19], and random search, on real-data benchmarks. In particular, we assessed the algorithms on some of the same benchmarks present in the original BORE paper [20].

### D.3.1 Algorithm settings

All versions of BORE were set with $\gamma := 0.25$. The original BORE algorithm used a 2-layer, 32-unit fully connected neural network as a classifier. The network was trained via stochastic gradient descent using Adam [21]. As in the other experiments in this paper, we followed the same scheme that keeps the number of gradient steps per epoch fixed [see 20, Appendix J.3], set in our case as 100, and a mini-batch of size 64. The probabilistic least-squares version of BORE and BORE++ were configured with a GP classifier using the rational quadratic kernel [22, Ch. 4] with output scaling and independent length scales per input dimension. All GP-based algorithms used the same type of kernel. GP hyper-parameters were estimated by maximising the GP's marginal likelihood at each iteration using BoTorch's hyper-parameter estimation methods, which apply L-BFGS by default [17]. BORE++ was set with a fixed value for its parameter $\beta_t := 3$, the regularisation factor was set as $\lambda := 0.025$. Acquisition function optimisation was run for 500 to 1000 iterations via L-BFGS with multiple restarts using SciPy's toolkit [23]. Lastly, for the experiment with the MNIST dataset, we also used the Tree-structured Parzen Estimator (TPE) by Bergstra et al. [19] set with default settings from the HyperOpt package. All algorithms were run for a given number of independent trials and results are presented with their 95% confidence intervals[2]

### D.3.2 Benchmarks

**Neural network hyper-parameter tuning.** We first considered two of the neural network (NN) tuning problems found in Tiao et al. [20], where a two-layer feed-forward NN is trained for regression. The NN is trained for 100 epochs with the ADAM optimizer [21], and the objective is the validation mean-squared error (MSE). The hyper-parameters are the initial learning rate, learning rate schedule, batch size, along with the layer-specific widths, activations, and dropout rates. In particular, we considered Parkinson's telemonitoring [24] and the CT slice localisation [25] datasets, available at UCI's machine learning repository [26], and utilize HPOBench [27], which tabulates, for each dataset, the MSEs resulting from all possible (62,208) configurations. The datasets and code are publicly available[3]. Each algorithm was run for 500 iterations across 10 independent trials.

---

[2]Confidence intervals are calculated via linear interpolation when the number of trials is small.

[3]Tabular benchmarks: `https://github.com/automl/nas_benchmarks`

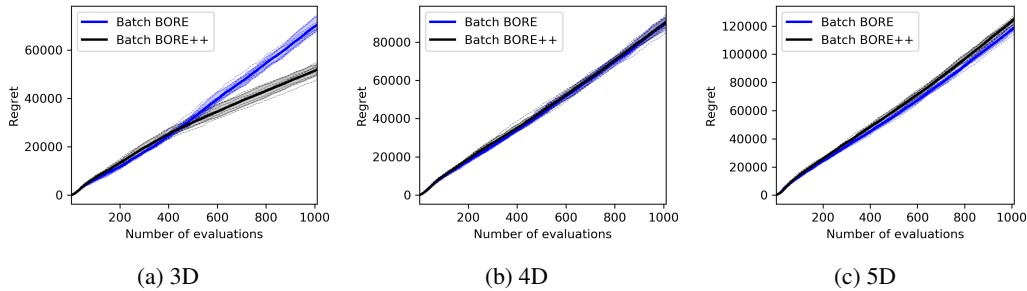

| (a) 3D | (b) 4D | (c) 5D |

Figure E.1: BORE vs. BORE++ in the batch setting tested on the Rosenbrock function at varying search space dimensionalities. The plots compare the cumulative regret of each algorithm averaged over 10 runs. Shaded areas correspond to the 95% confidence interval.

**Racing line optimisation.**  We compare the different versions of BORE against a random search baseline in the UC Berkeley racing line optimisation benchmark [28] using the code provided by Jain and Morari [29]. The task consists of finding the optimal racing line across a given track by optimising the positions of a set of 10 waypoints on the Cartesian plane along the track's centre line which would reduce the lap time for a given car, resulting in a 20-dimensional problem. For this track, the car model is based on UC Berkeley's 1:10 scale miniature racing car open source model[4]. Each algorithm was run for 50 iterations across 5 independent trials.

**Neural architecture search.**  Lastly, we compare all algorithms on a neural network architecture search problem. The task consists of optimising hyper-parameters which control the training process (initial learning rate, batch size, dropout, exponential decay factor for learning rate) and the architecture (number of layers and units per layer) of a feed forward neural network on the MNIST hand-written digits classification task [30]. The objective is to minimise the NN classification error. To allow for a wide range of hyper-parameter evaluations, this task uses a random forest surrogate trained with data obtained by training the actual NNs on MNIST [31]. For this experiment, each method was run for 200 iterations across 10 independent trials.

## E  Dimensionality effect on batch BORE vs. batch BORE++ performance

We compared batch BORE against the batch BORE++ algorithm on a synthetic optimisation problem with the Rosenbrock function. The dimensionality of the search space was varied. The cumulative regret curves for each algorithm are presented in Figure E.1.

Both algorithms were configured with a Bayesian logistic regression classifier applying random Fourier features [32] as feature maps based on the squared-exponential kernel. The number of features was set as 300, and the classifier was trained via expectation maximisation. Observations were corrupted by additive Gaussian noise with zero mean and a small noise variance $\sigma_\epsilon^2 = 10^{-4}$, and each model was set accordingly. To demonstrate the practicality of the method, the UCB parameter for BORE++ was fixed at $\beta_t := 3$ across all iterations $t \geq 1$, instead of applying the theoretical setup. SVGD was configured as its second-order version [33] applying L-BFGS to adjust its steps [14].

As the results show in Figure E.1, batch BORE++ has a clear advantage over batch BORE in low dimensions. However, the performance gains become less obvious at higher dimensionalities and eventually deteriorate. One of the factors explaining this behaviour is that, as the dimensionality increases, uncertainty estimates become less useful. Distances between data points largely increase and affect the posterior variance estimates provided by translation-invariant kernels, such as the squared-exponential kernel our feature maps were based on. Other classification models may lead to different behaviours, and their investigation is left for future work.

---

[4]Open source race car: https://github.com/MPC-Berkeley/barc/tree/devel-ugo