# OpenReview forum: "Batch Bayesian optimisation via density-ratio estimation with guarantees"
_NeurIPS.cc/2022/Conference — NeurIPS 2022 Accept_

### Official Review · Reviewer_wzi1 · 2022-07-07

**Rating:** 4
**Confidence:** 3
**Soundness:** 3 good
**Presentation:** 4 excellent
**Contribution:** 2 fair

**Summary:**

This paper presents BORE++ which improve uncertainty quantification over BORE. They present theoretical guarantees for the regret under certain conditions.

**Questions:**

Is BORE Bayesian? I understand that BORE follows the same sequential updating scheme as Bayesian optimisation. I struggle to see there is any prior in algorithm 1. So is BORE actually Bayesian or is it just probabilistic?

Following up on this question, if there is no reversion to the prior: how does BORE ensure a trade-off between exploitation and exploration? Question out of curiosity. (I see you discuss this in lines 121-123 too)

"Bayesian neural networks [42], random forests [43], dropout methods, etc. [44], also constitute valid approaches for predictive uncertainty estimation." Can you elaborate what valid means here?


**Limitations:**

There are no immediate societal impacts to be discussed in this paper.

**Strengths And Weaknesses:**

The related work section is short and should be more elaborate. The authors could consider placing it after the background section, where it should be easier to elaborate pointing to that section.

Theorem 1 is for pi which is not a valid classifier. Can we say anything about the case where pi optimal with the cross-entropy loss?

How should I relate the bounds in Theorem 2 to GP-UCB with some kernels, when BORE does not have any kernel choice? Are these new bounds for BORE (with no kernel), or just the "usual" GP-UCB bounds in a new language?

Can the authors explain that GP-UCB and BORE++ are fundamentally different? Phrased another way: if we use PLS classifer (as in 5.2), are they two parameterizations of the same mechanism?

The authors explain that BORE can be more flexible because no kernel or GP is needed. From my reading all theoretical guarantees assume a kernel on pi.

Figure 1 needs more ticks on y-axis.

In general, the experimental section of this paper does not allow any conclusion to be drawn from it. It lacks in both width and depth.

This is a theoretical paper, but I am to be convinced by the authors that the contribution is substantial to the literature. Specifically, the authors should clearly state that the conditions they prove under are different from GP-UCB.

---

> ### Author Response · Authors · 2022-07-29
> **Response**
>
> Dear Reviewer,
>
> We appreciate your comments and constructive criticism. In the following, we start addressing some of the highlighted issues.
>
> * ``The related work section is short and should be more elaborate.`` We agree that our related work section is relatively short, though we tried to be objective and succinct to save room for the theoretical derivations. We will try to expand the section with other relevant related work and reposition it, as suggested. We are open to suggestions.
> * ``Theorem 1 is for pi which is not a valid classifier. Can we say anything about the case where pi optimal with the cross-entropy loss?``
>     - We are not sure if the Reviewer refers to $\pi$ or $\hat\pi$. The latter is usually not a valid probabilistic classifier, as its outputs may fall outside the [0,1] range. However, Theorem 1 is stating that the approximation error of $\hat\pi$ with respect to the true classifier $\pi$ is bounded with high probability by a quantity that can be calculated if one knows the reproducing kernel of the Hilbert space where the true classifier lies in.
>     - Working with the cross-entropy loss is objective of future work. This loss is not as amenable to deal with due to its intrinsic non-linearities. One approach is to try to linearise it (via, e.g., Taylor expansions), combined with a suitable formulation for $\pi(\mathbf x)$, and then bound the linearisation error in terms, for example, of upper bounds on derivatives. One would then have to come up with possibly novel theoretical results bounding the pointwise approximation error of the classifier in this setting under BO's *non-i.i.d.* (and *non-stationary*) data-generating process to be able to analyse the theoretical regret. The resulting regret bound, if achievable, however, would possibly have a very different and harder-to-interpret form when compared to the results we provide. We will comment on these issues within a dedicated section in the revised Appendix.
> * ``How should I relate the bounds in Theorem 2 to GP-UCB with some kernels, when BORE does not have any kernel choice? Are these new bounds for BORE (with no kernel), or just the "usual" GP-UCB bounds in a new language?`` Theorem 2 addresses BORE with a probabilistic least-squares (PLS) classifier assumed to be a member of an RKHS. In this case, the optimal least-squares classifier has an associated uncertainty estimate given by $\sigma_t$ (see Theorem 1), which is not used by BORE, but is quantifiable for the theoretical results. Note that the "optimal least-squares classifier" can be considered as an ideal classifier which any (stochastic) gradient descent algorithm training a kernel-free parametric model (e.g., neural network) would be trying to approach. Therefore, the result in Theorem 2 applies to BORE (even with no kernel) and basically tells us that, in this idealised, fully-converged setting, if the algorithm never collects a sample at the optimum, the regret does not vanish. As explained in the paragraph just below Theorem 2, it could be the case that, if the first observation is not at the optimum, the algorithm simply gets stuck at the same location, while never observing the optimum. However, in practice, BORE would work with stochastic approximations to the optimal classifier, and the behaviour turns out different, possibly closer to the case of Thompson sampling (see Appendix B for a discussion).
>
> * ``Can the authors explain that GP-UCB and BORE++ are fundamentally different? Phrased another way: if we use PLS classifer (as in 5.2), are they two parameterizations of the same mechanism?`` Yes, both algorithms are fundamentally different, though sharing a few similarities in the PLS-BORE setting. GP-UCB operates directly with upper confidence bounds over objective function values, while BORE++ uses a UCB on the probability of improvement over a threshold.
>
> * ``The authors explain that BORE can be more flexible because no kernel or GP is needed. From my reading all theoretical guarantees assume a kernel on pi.`` A kernel is only needed to estimate a quantile over the predictions. The RKHS formulation does not necessarily require a kernel machine. There are other forms of estimating quantiles over the classifier probability. In the RKHS least-squares setting, it only happens that quantiles are available in closed form. However, we can estimate quantiles that satisfy the condition in Equation 22 in other settings as well (e.g., via sample-based estimates and probabilistic union bounds). We will comment on that in the revision.
>
> We will address the specific questions and remaining points in a follow-up comment. Regarding experiments, we refer to the response to the other reviews. We will provide more updates shortly.
>
>
> Kind regards,
>
> The Authors

---

> > ### Author Response · Authors · 2022-08-02
> > **On Bayesian aspects**
> >
> > > Is BORE Bayesian? I understand that BORE follows the same sequential updating scheme as Bayesian optimisation. I struggle to see there is any prior in algorithm 1. So is BORE actually Bayesian or is it just probabilistic?
> >
> > Both of the points raised are actually addressed in some detail in the "Discussion and Outlook" section of the original BORE paper (ICML 2021 version), under the paragraph headings "Hyperparameter estimation" -- which discusses the potential benefits of a fully-Bayesian treatment in BORE with a prior placed on parameters, particularly as it relates to exploration, and also under heading "Exploration" which discusses the general tendency for BORE to under-explore, and provides suggestions for addressing these. This present work also discusses these issues and has a key aim of addressing the very same problems.
> >
> > > "Bayesian neural networks [42], random forests [43], dropout methods, etc. [44], also constitute valid approaches for predictive uncertainty estimation." Can you elaborate what valid means here?
> >
> > By "valid", we meant that these methods also quantify uncertainty in their predictions in a consistent way, i.e., the uncertainty reduces as the posterior distribution concentrates around the parameters which better explain the data as more observations become available.

---

> > > ### Comment · Reviewer_wzi1 · 2022-08-09
> > > **Thank you**
> > >
> > > I would like to thank the reviewers for their responses. I am still on the border: can the reviewers say which experiments they will add to the paper with the extra page, and briefly conclude on what these experiments show?
> > >
> > > My concern with this paper is still the experimental evaluation, which I feel a paper like this needs.

---

> > > > ### Author Response · Authors · 2022-08-09
> > > > **Experiments**
> > > >
> > > > Thanks for the feedback. We are planning to add the experimental results in Sec. E.2 in the appendix, which were run on real-data baselines, to the main paper in the extra page, case the paper gets accepted.

---

### Official Review · Reviewer_ect9 · 2022-07-11

**Rating:** 6
**Confidence:** 4
**Soundness:** 3 good
**Presentation:** 3 good
**Contribution:** 3 good

**Summary:**

The paper provides a regret analysis of previous work which re-formulated Bayesian optimization with expected improvement as optimization over the input space of a discriminative classifier (BORE). Assuming the discriminator $p(y \le \tau | x)$ is a member of a reproducing kernel Hilbert space (RKHS) allows modeling the discriminator using kernel least squares which in turn yields a high probability bound on the discriminator error. From this, bounds on instantaneous and cumulative regret follow. The paper then addresses BORE's potential for a non-vanishing cumulative regret by proposing a UCB-like variant BORE++ which achieves vanishing regret. Then, a batch variant of BORE++ utilizing Stein variational gradient descent is introduced, along with regret analysis, and synthetic and real-data experiments are conducted.

**Questions:**

* Does the result of theorem 3 only apply when the true $\pi(x)$ is in the RKHS associated with the least-squares kernel?

* How does the performance of BORE++ compare on standard benchmarks to other non-batch BO methods? Particularly, how does it compare to BORE wrt/ the experiments included in the original BORE paper?

* The batch BORE++ guarantee relies on the Lipschitz constant $L_\pi$ being bounded away from zero. However, this appears at odds with a decreasing $\tau$ value to better identify the global minimum. Is there an adaptive schedule for $\tau$ that can retain vanishing regret while locating the optimum?

**Limitations:**

There wasn't any discussion that I saw on the limitations of the BORE++ variants or potential negative societal impacts. For the former, some qualitative discussion of which black-box optimization settings BORE++ is preferable / not preferable to standard BO with EI would help.

**Strengths And Weaknesses:**

Originality

* The paper mostly appears to utilize previous results in Gaussian process-based Bayesian optimization, online learning, and an approximate inference guarantee in its development. In this sense, the technical contributions seem somewhat incremental.

Quality

* The experimental setup oddly doesn't include non-batch BORE++ which is a primary contribution of the paper.

* For a batch method, I was a little underwhelmed by the experimental results. However, I do want to commend the authors for including two cases where the proposed method isn't the clear winner.

Clarity

* The paper was generally very clear and easy to follow.

* The authors might consider moving the SVGD description (sec 3.4) to the description of the batch alg (sec. 6) since it is not relevant prior to this point.

Significance

* Had the experimental results been more thorough and clearer, the paper's impact might have been higher. Given that BORE is fairly recent and not very prevalent as a standard tool in BO, the significance of guarantees is maybe a question.

---

> ### Author Response · Authors · 2022-07-29
> **Review response**
>
> Dear Reviewer,
>
> We appreciate your comments and insightful feedback. We will start addressing some of the issues highlighted by the review.
>
> As mentioned in the response to Reviewer *tUys*, we agree that the experimental evaluation in the paper is somewhat limited. However, we would like to emphasise that the focus of the paper is on the theoretical analysis. The experiments were mainly meant to demonstrate the differences in terms of practical performance between the proposed methods and their baselines, and not meant to establish a new state-of-the-art, which would require a much more extensive evaluation. We will clarify this point in the revision. In any case, we agree that the paper is lacking a comparison between BORE++ and BORE in the sequential/non-batch setting, along with other standard BO baselines other than expected improvement, which is assessed in the experiments. We are preparing some preliminary experimental results to add to the revision and post in our response here during the rebuttal phase.
>
> #### Answer to questions:
> 1. *Does the result of theorem 3 only apply when the true kernel?*
>
>     - Yes, and we recognise this assumption should be stated or referred to in the theorem. We will fix that in the revision.
> 2. *How does the performance of BORE++ compare on standard benchmarks to other non-batch BO methods? Particularly, how does it compare to BORE wrt/ the experiments included in the original BORE paper?*
>
>     - We will address this with the upcoming results.
> 3. *The batch BORE++ guarantee relies on the Lipschitz constant $L_\pi$ being bounded away from zero.*
>     - (a) *However, this appears at odds with a decreasing $\tau$ value to better identify the global minimum.*
>
>         - As $\tau$ approaches the true observations quantile, more and more regions of the search space will have a true classifier probability close to 0. However, as the observations are noisy, and the conditions in Theorem 2 guarantee the noise distribution to have support over the entire real line, due to the strict monotonicity assumption, there will always be some probability, even if very small, that an observation will fall under the threshold due to noise. Recall that $\tau$ is defined as the noisy observations quantile, instead of directly over the true objective function.
>     - (b) *Is there an adaptive schedule for $\tau$ that can retain vanishing regret while locating the optimum?*
>
>         - Currently, in the practical implementation of BORE, $\tau$ is adapted over time, as it is estimated based on the empirical observations distribution. This adaptation scheme was not considered in our theoretical results as it violates some of the main assumptions of the GP-UCB theory we applied for the analysis of BORE (see Appendix C for a discussion). If the adaptation scheme, however, was deterministic, i.e., not depending on the observations, such as annealing an initial estimate of $\tau$ (e.g., based on random samples) according to a deterministic schedule, then we believe our results would remain valid, since this setting would not affect the filtration of the data-generating stochastic process we analysed. This construction might lead to other interesting theoretical versions of BORE which could be objective of future work.
>
> Kind regards,
> The Authors

---

> > ### Author Response · Authors · 2022-08-01
> > **On the significance of the results**
> >
> > > Given that BORE is fairly recent and not very prevalent as a standard tool in BO, the significance of guarantees is maybe a question.
> >
> > Regarding the significance of the results, as usual, it is difficult to predict the long-term impact of any piece of scientific work, especially early on. However, there are some good reasons to expect BORE to gain widespread adoption and have a significant impact in the near future.
> >
> > Firstly, we address the real-world use of BORE. Despite the introduction of many players in the space in recent years, toolkits such as HyperOpt (https://github.com/hyperopt/hyperopt) and Optuna (https://github.com/optuna/optuna) still remain the most widely used for hyper-parameter optimisation, particularly in the domain of AutoML. These libraries rely foremost on the Tree-structured Parzen Estimator (TPE) density estimation method by Bergstra et al. [2011] as their default search algorithm. In many settings, e.g. those of high-dimensionality, TPE consistently outperforms other paradigms such as evolutionary strategies or traditional GP-based BO. By directly addressing several profound shortcomings of TPE, BORE has been proven to, in turn, consistently outperform TPE. This was not only demonstrated in the original paper but has been independently observed in subsequent works. Thus seen, BORE stands poised to be an ideal candidate to replace TPE and become a predominant method for hyper-parameter search. In fact, BORE has already been adopted as one of the primary search algorithms in Syne Tune [Salinas et al. 2022], a rapidly growing open-source framework for hyper-parameter optimisation built by Amazon Research.
> >
> > Secondly, we discuss BORE as a topic of research. Less than a year on from its initial publication, it already features in the upcoming textbook on Bayesian Optimisation by Roman Garnett [2022], due to be published later this year. Additionally, several prominent research labs have already made efforts to extend BORE in important ways, such as generalising it to other acquisition functions [Song et al. 2022] or to multiple objectives [De Ath et al. 2022].
> >
> > In like manner, this work seeks to build upon BORE by bridging the gap between practice and theory. We hope that providing guarantees and a deeper understanding from a theoretical perspective will further increase the potential impact of BORE.
> >
> > ### References:
> > - Bergstra, James, Remi Bardenet, Yoshua Bengio, and Balazs Kegl. (2011). “Algorithms for Hyper-Parameter Optimization.” In *Advances in Neural Information Processing Systems (NIPS),* 2546–54.
> >
> > - De Ath, G., Chugh, T., & Rahat, A. A. M. (2022). MBORE: Multi-Objective Bayesian Optimisation by Density-Ratio Estimation. In *Proceedings of the Genetic and Evolutionary Computation Conference (GECCO'22),* 776–785.
> >
> > - Garnett, R. (2022). *Bayesian Optimization.* Cambridge University Press. [In preparation] https://bayesoptbook.com/
> >
> > - Salinas, D., Seeger, M., Klein, A., Perrone, V., Wistuba, M., & Archambeau, C. (2022). Syne Tune: A Library for Large Scale Hyperparameter Tuning and Reproducible Research. In *First Conference on Automated Machine Learning (AutoML-Conf 2022)* [Main Track].
> >
> > - Song, J., Yu, L., Neiswanger, W., & Ermon, S. (2022). A General Recipe for Likelihood-free Bayesian Optimization. In *Proceedings of the 39th International Conference on Machine Learning (ICML).* Baltimore, Maryland, USA: PMLR, volume 162.

---

### Official Review · Reviewer_HW2t · 2022-07-12

**Rating:** 5
**Confidence:** 2
**Soundness:** 3 good
**Presentation:** 3 good
**Contribution:** 3 good

**Summary:**

The paper proposes an extension of BORE, BO based on the direct density-ratio estimation, and its regret analysis is also provided. The authors first derived a regret bound for the original BORE and mentioned a problematic term remains. Based on this observation, a UCB based extension, called BORE++, is proposed, and the authors revealed BORE++ has the same good regret convergence as the original GP-UCB.

**Questions:**

[1] In BORE++, how is \sigma_t(x) in (23) estimated? It is seems described in 5.1 (empirical quantile), but I missed, in the end, which uncertainty estimation is employed, and how it affects the theoretical guarantee?

[2]
> Therefore, batch BORE++ should be able to achieve better performance than BORE++ while running the same number of function evaluations.

This is a bit counter-intuitive for me. In general, for each one of decision makings, the non-batch model can use a larger number of observations (The batch model can only use t observations during determining next M observations, while the non-batch model can gradually increase observations (e.g., to observe M-th observation, it can use t+(M-1) observations)). Therefore, the claim that the batch model achieve better performance than the non-batch model by the same number of observations is a bit surprising for me. Could you explain its rationale in more detail?

**Limitations:**

I could not find clear descriptions about current limitation.

**Strengths And Weaknesses:**

The paper is well-organized and easy to follow regardless of its technical complexity.

Overall, I think that the regret analysis of this paper is informative for the community. The consequence is interesting (BORE has a problematic term and BORE++ can avoid it).

Empirical evaluation is a bit weak. It only shows the batch setting, though BORE++ would be a novel acquisition function even for the non-batch setting. More comprehensive evaluation with a standard acquisition functions would have been convincing.

---

> ### Author Response · Authors · 2022-07-29
> **Review response**
>
> Dear Reviewer,
>
> We appreciate your comments and insightful feedback. We will start addressing some of the issues highlighted by the review.
>
> As mentioned in the response to Reviewer *tUys*, we agree that the experimental evaluation in the paper is somewhat limited. However, the focus of the paper is on the theoretical analysis. The experiments were mainly meant to demonstrate the differences in terms of practical performance between the proposed methods and their baselines, not to establish a new state-of-the-art, which would require a much more extensive evaluation. We will clarify this point in the revision. In any case, we agree that the paper is lacking a comparison between BORE++ and BORE in the sequential/non-batch setting, along with other standard BO baselines other than expected improvement, which is assessed in the experiments. We are preparing some preliminary experimental results to add to the revision and post in our response here during the rebuttal phase.
>
> Regarding your specific questions, the formulation of $\sigma_t$ is described in Theorem 1, though we understand that it is not obvious in the other theoretical results. We will add more emphasis to it in the revision, perhaps giving $\sigma_t$ its own dedicated equation, depending on rearrangements of the contents, since it is an important variable for the implementation of BORE++ and its batch version in the PLS setting.
>
> About the theoretical performance differences between batch BORE++ and its sequential version, we understand that the quoted statement is counter-intuitive and possibly confusing. What happens is that Theorem 4 is addressing the "expected" regret per "batch iteration", which does not correspond to the total number of observations collected up to a point in time, as the instant regret in Theorem 3 would. However, Theorem 4 is considering that function evaluations can occur in parallel at each iteration, so that the expected regret becomes a more appropriate approach to understand the advantages of the batch algorithm. The main performance gain is, therefore, in terms of the regret accumulation over time for the same number of evaluations. In accordance with the Reviewer's reasoning, a comparison in terms of instant regret would actually lead to a worse regret bound for the batch algorithm. The simplest way to see that would be to compare the average regret, since the expected regret of each of the $M$ samples within a batch is the same. After collecting $M$ observations, the sequential algorithm would have an average instant regret of a factor of $\mathcal{O}(\xi_M/\sqrt{M})$ times that of the batch algorithm, factor which vanishes as $M\to\infty$ for most popular kernels. In essence, any sequential algorithm would have an advantage in this sense over a batch-based counter-part, since the sequential algorithm has access to more information per sample selection step than the batch algorithm. With that in mind, we realise that the quoted statement is not specific regarding the type of performance gain we refer to and will adjust it accordingly in the revision.
>
> Kind regards,
> The Authors

---

### Official Review · Reviewer_tUys · 2022-07-13

**Rating:** 7
**Confidence:** 4
**Soundness:** 3 good
**Presentation:** 4 excellent
**Contribution:** 3 good

**Summary:**

The paper targets multiple problems. First, it develops the theoretical analysis (in particular the regret analysis) for the existing BORE framework by Tiao et al [9], helping to understand BORE's convergence property. Second, it addresses the non-convergence issue of BORE by proposing the method BORE++ which improves the uncertainty estimates combining with using an upper confidence bound (UCB) on the predicted class probabilities as the acquisition function to improve the convergence property. Finally, the paper proposes the batch BORE method along with its theoretical convergence analysis. Some numerical experiments are conducted to assess the theory developed and to compare the performance of the proposed batch BORE method with existing baselines.

**Questions:**

Apart from some comments mentioned in the Weaknesses section which the authors can response, I also have the additional following questions:

•	In Section 7.1, Theory Assessment, when assessing the theoretical results of the proposed method BORE, BORE++ and GP-UCB, I'm just wondering what are the values of some parameters in these algorithms such as beta_t, L_{epsilon}, etc.? I saw that in the appendix, the experiment setup is described in detail, however, I couldn't find the information regarding some parameters listed in the previous sentence.

•	I would love to understand more about the performance of BORE++ versus BORE, in particularly for more objective functions.


**Limitations:**

Yes, the limitations are described.

**Strengths And Weaknesses:**

Strengths:

•	The paper is very well-written and clear. The proposed methods, the theoretical analysis and the discussion regarding the theoretical analysis are very well presented.

•	The theoretical analysis for the BORE framework, the BORE++ method and the batch BORE method is very interesting. To the best of my knowledge, the analysis seems to be correct. The techniques used are new and can inspire future research works. I know these are difficult problems to perform theoretical analysis. I also like the fact that the paper tries to explain the impact of the theoretical analysis and discuss some assumptions used in the theoretical analysis.

•	Although limited, I also like that the paper tries to perform an experiment to assess the theoretical analysis.


Weaknesses: In general, to me, the experiments are a bit weak/limited. Please find the detailed comments as follows.

•	Experiments are only conducted to compare the proposed Batch BORE method with existing baselines, and very limited experiments are conducted to compare the performance of BORE++ over baselines (e.g., BORE or GP-UCB).

•	For the experiments that compare the proposed Batch BORE method with existing baselines, only synthetic objective functions are used, there are no real-world problems.

---

> ### Author Response · Authors · 2022-07-29
> **Response to review**
>
> Dear Reviewer,
>
> We appreciate your comments and insightful feedback. We will start addressing some of the issues highlighted by the review.
>
> We agree that the experiments section in the paper is somewhat limited. However, as the reviewer has noticed, the focus of the paper is on the theoretical analysis. The experiments were mainly meant to demonstrate the differences in terms of practical performance between the proposed methods and their baselines, not to establish a new state-of-the-art, which would require a much more extensive evaluation. In any case, we agree that a comparison between BORE++ and BORE in the sequential/non-batch setting is lacking, and we are preparing some preliminary experimental results to add to the revision and post in our response here during the rebuttal period.
>
> Regarding your questions about the parameters settings for the theory assessment experiments, $\beta_t$ was set according to Theorem 1, which requires a confidence level $\delta \in  (0, 1)$, an estimate (or upper bound) for the norm of the true classifier in the RKHS, i.e., $\lVert \pi \rVert_k$, and the value for the L2-regularisation parameter $\lambda$ (used as GP noise variance parameter). We noticed that these settings might be a bit hard to find in the midst of Sec. D.1 in the appendix, and, for instance, $\lambda := 0.025$ was not explicitly mentioned to be the setting for BORE++. We'll revise Sec. D.1 to make these settings easier to find (possibly in a table) in the revised paper.
>
> The value of $L_\epsilon$ is not needed for the algorithms. It only appears as a quantity that contributes to the theoretical regret bounds, but it does not need to be provided to the algorithms.
>
> Kind regards,
> The Authors

---

> > ### Comment · Reviewer_tUys · 2022-08-06
> > **Thank you for the authors' response**
> >
> > Dear authors,
> >
> > Thank you for your response. I appreciate the authors putting more effort on running experiments to understand the performance of BORE and BORE ++. My judgement and opinion about the paper remain positive. I still do think the paper does have some contribution to the community, therefore, I still keep my score as is, which is 7.

---

### Author Response · Authors · 2022-08-02
**Experiments and revisions**

Dear Reviewers,

We once again would like to thank you all for your reviews and the constructive feedback that was received. As mentioned in our individual responses, new preliminary results comparing BORE++ against BORE and other baselines in the sequential/non-batch setting on real-data benchmarks have been included in the revision. The results can be found in Appendix E.2 in the supplement. We assessed the algorithms on well known *neural network tuning*, *neural architecture search* and *racing line optimisation* problems. In summary, the results show that BORE++, although using a different loss function and a simple GP-based classifier, is able to present better or similar performance when compared with the original BORE using a multi-layer perceptron model. In all cases, BORE++ significantly outperforms the probabilistic least-squares (PLS) version of BORE, also implemented with a GP, representing an optimally trained, minimal-loss classifier, confirming the theoretical results. In fact, it's possible to note that BORE (PLS) performs comparably to (or at times worse than) random search, which was also included in the results, indicating that the least-squares classifier is unable to properly capture epistemic uncertainty.

We also added a few revisions to both the main paper and the supplement, taking into account some of your reviews. The revised text is highlighted with a different colour, so that it can be easily identified. The highlights will, of course, be removed in the published version, case the paper gets accepted. Due to time constraints, we were not able to address every issue in the revision, as we focused on the new experimental results. We are, however, keeping track of the issues to later address them. In any case, we are committed to continue working towards improving the paper and its presentation, following up on the feedback from the reviews and the upcoming discussions.


Kind regards,

The Authors

---

> ### Author Response · Authors · 2022-08-09
> **Additional baselines**
>
> Dear Reviewers,
>
> We have added additional baselines to the experimental results in Appendix E.2 (supplement), which now include standard BO baselines: GP-UCB and GP-EI, and have updated the discussion of these results. The main conclusions, however, remain that improved uncertainty estimates can lead to practical performance gains in the BORE framework setting. We have also revised the wording of the first paragraph in the main paper's Experiments section (Sec. 7) in an attempt to better state the purpose of the experimental evaluations.
>
>
> Best regards,
>
> The Authors

---

### Meta-Review · Area_Chair_E8vq · 2022-08-27

**Recommendation:** Accept
**Confidence:** Less certain

**Metareview:**

This paper provides a theoretical analysis of the regret for the BORE framework and addresses existing issues with a BORE++ method that provides uncertainty quantification in the estimation of the classifier. An analysis of BORE++ is also presented alongside an exploration of the batch Bayesian optimization setting.

Reviewers appreciated the theoretical analysis provided in the paper as well as the explanation of the theoretical results and the assumptions needed for those results to hold. Several reviewers expressed the main weaknesses were the empirical analyses in the paper, including the results focusing on the batch case. However, after the rebuttal and revision, several reviewers felt like the additional results were convincing enough to recommend acceptance for this work.

Please use the extra page to finish addressing the remaining comments of the reviewers. Some additional points to keep in mind include: 1) the related work section could be expanded to include more literature as well as the context in which the current work relates to the work cited; 2) the figures could be improved to be easier to read, e.g., by increasing font sizes.

**Award:**

No

---

### Decision · Program_Chairs · 2022-09-14

Accept